# Association between first-trimester cell-free fetal DNA levels and the risk of preterm birth and low birth weight: A propensity score-matched cohort study

Xiaosa Wang[1], Hui Shao[2], Junjiang Wu[1], Guofeng Yin [3]*

**1** Department of Obstetrics, Shaoxing Maternity and Child Health Care Hospital, Maternity and Child Health Care Affiliated Hospital, Shaoxing University, Shaoxing, China, **2** Department of infectology, Shaoxing Maternity and Child Health Care Hospital, Maternity and Child Health Care Affiliated Hospital, Shaoxing University, Shaoxing, China, **3** Department of Pediatrics, Shaoxing Maternity and Child Health Care Hospital, Maternity and Child Health Care Affiliated Hospital, Shaoxing University, Shaoxing, China

* ygfeng1985@163.com

## Abstract

### Objective

To investigate the association between the concentration of cell-free fetal DNA (cffDNA) in the first trimester and the risks of preterm birth (PTB) and low birth weight (LBW) in a large cohort.

### Methods

This study employed a propensity score matching (PSM) analysis to balance baseline characteristics between case and control groups. Univariate and multivariate logistic regression models were used to assess the association between cffDNA (analyzed in quartiles and as a binary variable) and the outcomes. Hierarchical regression models were built with progressive adjustment for confounders. Subgroup and threshold effect analyses were conducted to evaluate the consistency and nature of the associations.

### Results

Before PSM, significant baseline differences were observed across multiple covariates. After 1:3 ratio PSM, 682 PTB cases were matched with 1,983 controls and 533 LBW cases with 1,579 controls, achieving statistical balance. For PTB, a consistent, significant inverse association was observed across all cffDNA quartiles. Higher cffDNA levels were associated with a reduced risk of PTB in all models, both before and after matching. For LBW, a significantinverse association was confined to the highest cffDNA quartile (Q4), indicating a threshold effect. Subgroup analyses confirmed the robustness of these associations across most demographic strata.

**Data availability statement:** All relevant data are within the paper and its Supporting Information files.

**Funding:** The author(s) received no specific funding for this work.

**Competing interests:** The authors have declared that no competing interests exist.

Threshold analysis identified a significant inflection point for PTB risk both before and after PSM.

## Conclusion

Elevated cffDNA levels are significantly associated with a reduced risk of adverse pregnancy outcomes, but the relationship patterns differ. For preterm birth, the protection is graded across all cffDNA levels, while for low birth weight, it is a threshold effect evident only at the highest concentrations. These findings suggest cffDNA may serve as a biomarker of placental health and a useful tool for risk stratification.

## Introduction

Preterm birth (PTB), defined as delivery before 37 weeks of gestation, and low birth weight (LBW), defined as a birth weight below 2500 grams, represent major challenges in perinatal medicine and contribute significantly to global neonatal mortality and long-term morbidity [1–2]. Spontaneous preterm birth, the predominant subtype of PTB, has a multifactorial etiology, with previous studies identifying placental dysfunction, intrauterine inflammation, and infection as key pathophysiological drivers [3–4]. Similarly, LBW, whether occurring in conjunction with PTB or resulting from independent intrauterine growth restriction, is frequently associated with placental insufficiency leading to impaired nutrient and oxygen transfer [5]. Despite considerable research efforts, there remains a critical lack of robust early-pregnancy biomarkers capable of accurately identifying women at high risk for spontaneous PTB and LBW, which significantly hampers the development and implementation of targeted preventive strategies.

The advent of non-invasive prenatal testing (NIPT) has established cell-free fetal DNA (cffDNA) as a cornerstone of modern prenatal care, providing a unique window into placental health. cffDNA, predominantly derived from apoptotic trophoblasts, is detectable in maternal plasma during early pregnancy [6–7]. Its concentration is dynamic, influenced by gestational age, placental mass, and integrity. Conventionally, elevated cffDNA levels have been interpreted as a marker of increased placental turnover, often resulting from pathological processes such as ischemia, oxidative stress, or inflammation [8]. Numerous studies have confirmed that abnormal cffDNA levels in maternal plasma during the first and second trimesters are significantly associated with an increased risk of subsequent placental disorders, as robustly validated in conditions like preeclampsia [9].

However, when the research focus shifts to preterm birth and low birth weight, the existing evidence presents notable contradictions. While some studies support the view that elevated cffDNA levels signal increased risk, other influential investigations have reported divergent findings. For instance, a prospective cohort study found no significant association between first-trimester cffDNA levels (fetal fraction) and spontaneous preterm birth [10]. Furthermore, more refined analyses have suggested that it is not necessarily elevated but potentially reduced cffDNA levels that are

associated with an increased risk of spontaneous preterm birth [11]. This inconsistency underscores a critical knowledge gap: whether a relative deficiency in cffDNA could also be a marker of placental insufficiency, and whether its association with PTB differs from that with LBW. Large-scale cohort studies are needed to investigate these potential outcome-specific association patterns.

It remains unclear whether first-trimester cffDNA levels can serve as a biomarker for a risk-stratified approach to predicting adverse pregnancy outcomes. To address this question and to clarify its specific role, this study was designed to investigate the associations between cffDNA levels and the risks of PTB and LBW. We utilized a large cohort of over 10,000 singleton pregnancies, categorizing cffDNA into quartiles to enable risk stratification and employing multiple logistic regression models adjusted for key confounders. We specifically aimed to determine if cffDNA is an independent risk factor, to elucidate whether it has outcome-specific association patterns, and to examine if its effects on PTB and LBW are consistent or divergent.

## Methods

### Date of access to data

Data accessed on 1 May 2025 and authors had access to information that could identify individual participants during or after data collection.

### The study population

The study included 10,345 singleton neonates and their mothers who delivered at our institution between June 2023 and June 2025.In our institution, non-invasive prenatal testing (NIPT) was offered to pregnant women meeting specific clinical criteria during the study period. These indications included: (1) women with first-trimester serum screening results indicating a risk for common fetal chromosomal aneuploidies between the high-risk cutoff and 1/1000; (2) women with contraindications to invasive prenatal diagnosis (e.g., threatened miscarriage, fever, bleeding tendencies, active chronic infectious diseases, or Rh-negative blood type); and (3) women who voluntarily requested risk assessment for trisomy 21, trisomy 18, and trisomy 13. Consequently, the study cohort represents a selected population of pregnant women who underwent NIPT based on these indications, rather than an unselected general obstetric population. Fig 1 presents the participant selection flowchart. The diagram details the process from initial eligibility assessment to final analytic sample inclusion. It also outlines the subsequent statistical approaches, including propensity score matching (PSM), threshold and subgroup analyses, and hierarchical logistic regression modeling.This retrospective study was approved by the Ethics Committee of Shaoxing Maternal and Child Health Hospital in January 2023, the ethical approval number is [IRB-AF-023–01.5]. And it did not involve animal or human clinical trials. And data collection was based on complete medical records and data analysis was performed anonymously. The ethics committee waived the requirement for informed consent. All our research methods were in accordance with relevant guidelines and regulations.

### Data collection

We conducted a retrospective analysis of routinely collected clinical data from electronic medical records and nursing care systems to examine basic maternal and neonatal characteristics. Cell-free fetal DNA (cffDNA) levels, expressed as fetal fraction (%), were quantitatively measured through clinical NIPT. The analysis was performed following standard protocols: maternal peripheral blood samples were collected, maternal plasma separation, DNA extraction, library preparation using the Illumina TruSeq Nano DNA Library Prep Kit,next-generation sequencing on the Illumina platform with a median sequencing depth of approximately 0.2X (10 million reads per sample), and bioinformatic quantification of the fetal DNA fraction. Fetal fraction was calculated using a validated algorithm that combines two approaches depending on fetal sex: for male fetuses, the algorithm is based on the relative coverage of chromosome Y sequences; for female fetuses, a

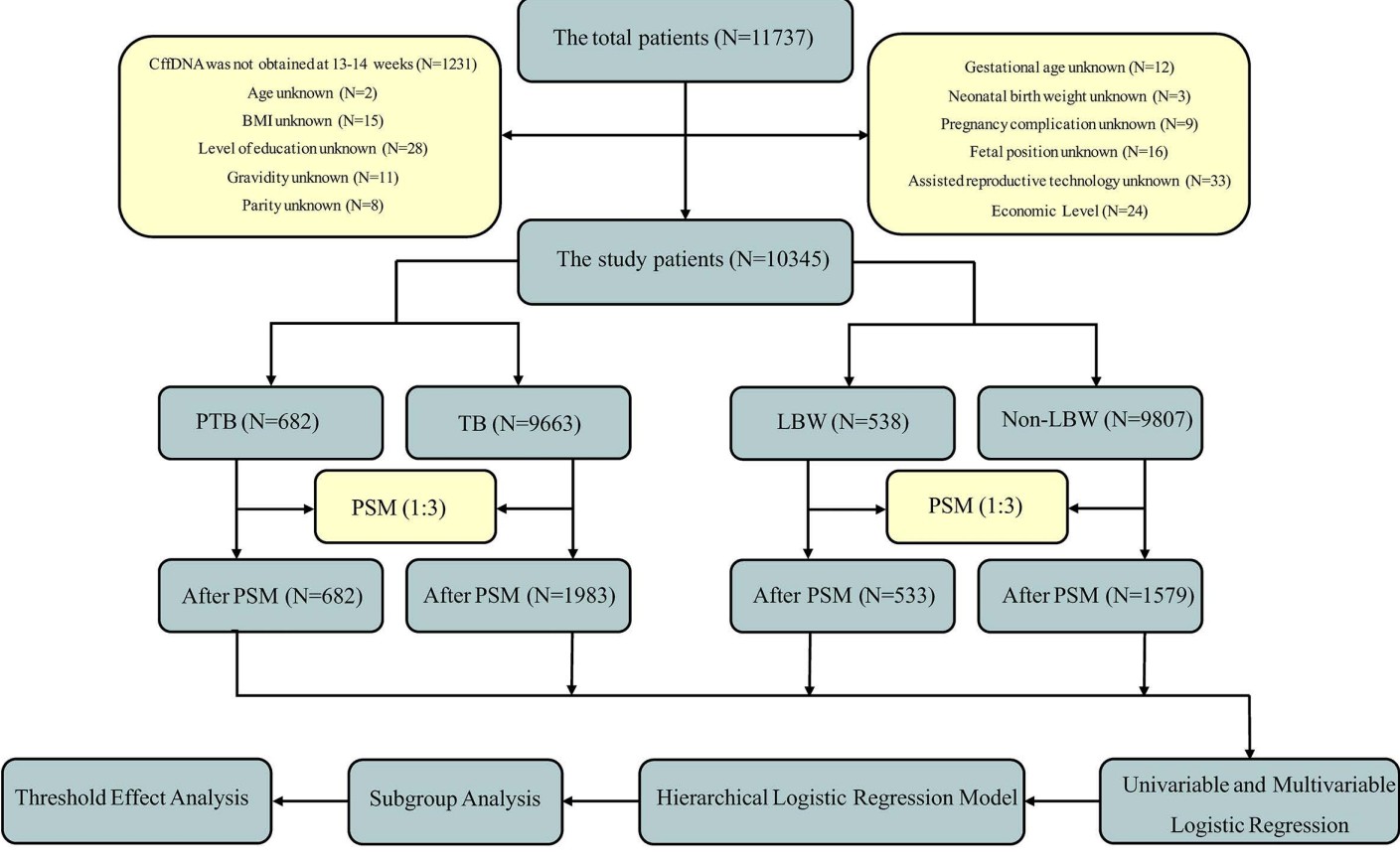

**Fig 1. Flowchart of participant screening and study inclusion process.**

neural network model trained on autosomal read count patterns and fragment size distributions is used to estimate fetal fraction. This algorithm is routinely applied in clinical NIPT reporting in our laboratory. A single, consistent analytical pipeline was maintained throughout the entire study period from June 2023 to June 2025, with no changes to the sequencing platform, library preparation protocol, or fetal fraction estimation algorithm. All samples were processed and analyzed using the same standardized protocols in our clinical laboratory. Quality control thresholds were applied to ensure reliable cffDNA measurement: only samples with a minimum sequencing depth of 10 million reads and a fetal fraction value ≥4% were included in the final analysis, as lower fetal fractions are associated with increased risk of test failure and may reflect poor sample quality or underlying placental abnormalities. Samples failing to meet these quality criteria were excluded from the study cohort. The extracted variables included maternal age, pre-delivery body mass index, educational level, gravidity, parity, neonatal sex, floating population status (defined as individuals moving from less developed inland regions to coastal developed areas for temporary residence and work), pregnancy complications (gestational diabetes, pregestational diabetes, gestational hypertension, and pregestational hypertension), fetal position (categorized as cephalic or non-cephalic), mode of conception (assisted reproductive technology versus spontaneous conception), family economic level (classified as above average for those admitted to special care wards and average/below otherwise), cffDNA levels at 13–14 weeks' gestation, preterm birth (<37 weeks), and low birth weight (<2500 grams). Maternal age was dichotomized at ≥35 years based on the widely accepted definition of advanced maternal age, which is associated with increased risks of adverse pregnancy outcomes including preterm birth and low birth weight [12]. Pre-delivery body mass index

(BMI) was categorized using a threshold of ≥30 kg/m², consistent with the definition of obesity as a well-established risk factor for pregnancy complications [13]. All variables were carefully reviewed and validated by a team of experienced senior obstetricians, midwives, and neonatologists to ensure data accuracy. Given the relatively small proportion of missing data, we employed a complete-case analysis approach by excluding records with any missing variables from the final analysis.

## Statistical analysis

A comprehensive descriptive analysis was conducted for all study participants. Continuous variables are presented as mean ± standard deviation (SD) and compared using t-tests, while categorical variables are expressed as percentages and compared using chi-square tests. To enhance statistical power and minimize bias in this observational study, we employed propensity score matching with a 1:3 nearest-neighbor algorithm (one case matched to three controls) and a caliper width of 0.2 [14–15]. The 1:3 matching ratio was selected to maximize statistical power while maintaining adequate balance between groups, as matching more controls to each case can increase the precision of treatment effect estimates without substantially increasing bias, particularly when the number of available controls is sufficient. After matching, standardized differences were used to assess balance between groups, with an absolute value below 10% indicating adequate balance of baseline characteristics.CffDNA was categorized into quartiles (Q1-Q4, with the first quartile Q1 as reference) for analysis. Both univariable and multivariable logistic regression analyses were performed, before and after propensity score matching, to evaluate the association between cffDNA and adverse neonatal outcomes (preterm birth and low birth weight). Hierarchical logistic regression models (Model 1, Model 2, Model 3) were further utilized to explore these associations. Subgroup analyses were stratified by maternal age, pre-delivery BMI, educational level, gravidity, parity, neonatal sex, floating population status, pregnancy complications, fetal position, mode of conception, and family economic level. Additionally, threshold analysis was conducted to identify critical cffDNA cutoff values and characterize dose-response relationships with preterm birth and low birth weight. All statistical tests were two-sided, with p-values <0.05 considered statistically significant.All subgroup analyses and threshold effect analyses were exploratory in nature and were not pre-specified in the study protocol. Given the multiple comparisons performed, these findings should be interpreted with caution and considered hypothesis-generating rather than confirmatory.

## Results

### Baseline characteristics of confounding variables before and after PSM in the study population

This study utilized propensity score matching (PSM) to control for potential confounding factors in examining the association between cell-free fetal DNA (cffDNA) and adverse pregnancy outcomes. Prior to matching, significant baseline differences (all p < 0.05) were observed between comparison groups across multiple covariates. As shown in Table 1, before PSM, women who experienced preterm birth (PTB) demonstrated significantly different profiles compared to those with term births, including higher proportions of advanced maternal age (≥35 years: 23.31% vs 16.93%), obesity (BMI ≥ 30: 27.27% vs 20.44%), lower educational attainment (less than high school: 24.78% vs 19.54%), higher gravidity (≥3: 34.02% vs 27.67%), increased floating population status (33.14% vs 26.68%), fewer pregnancy complications (24.34% vs 37.74%), more non-cephalic presentations (15.84% vs 4.19%), greater use of assisted reproductive technology (ART: 16.13% vs 6.89%), and lower economic status (average and below: 91.06% vs 82.21%).Similarly, Table 2 reveals that before matching, low birth weight (LBW) cases showed significant differences from normal birth weight controls, with higher rates of advanced maternal age (23.05% vs 17.04%), lower educational attainment (24.72% vs 19.62%), higher gravidity (≥3: 32.34% vs 27.86%), increased floating population status (31.78% vs 26.85%), fewer pregnancy complications (27.88% vs 37.35%), more non-cephalic presentations (16.54% vs 4.32%), greater ART utilization (13.75% vs 7.16%), and predominantly lower economic status (average and below: 92.75% vs 82.25%).Following 1:3 ratio matching,

**Table 1. Baseline Characteristics Before and After PSM for the Association Between CffDNA and PTB.**

| Variable | Before PSM | | | | After PSM | | | |
|---|---|---|---|---|---|---|---|---|
| | Total (n = 10345) | Term birth (n = 9663) | Preterm birth (n = 682) | P | Total (n = 2665) | Term birth (n = 1983) | Preterm birth (n = 682) | P |
| Age, n (%) | | | | <0.05 | | | | 0.244 |
| <35 | 8550 (82.65) | 8027 (83.07) | 523 (76.69) | | 2086 (78.27) | 1563 (78.82) | 523 (76.69) | |
| >=35 | 1795 (17.35) | 1636 (16.93) | 159 (23.31) | | 579 (21.73) | 420 (21.18) | 159 (23.31) | |
| BMI, n (%) | | | | <0.05 | | | | 0.685 |
| <30 | 8184 (79.11) | 7688 (79.56) | 496 (72.73) | | 1954 (73.32) | 1458 (73.52) | 496 (72.73) | |
| >=30 | 2161 (20.89) | 1975 (20.44) | 186 (27.27) | | 711 (26.68) | 525 (26.48) | 186 (27.27) | |
| Level of Education, n (%) | | | | <0.05 | | | | 0.945 |
| Less than high school | 2057 (19.88) | 1888 (19.54) | 169 (24.78) | | 663 (24.88) | 494 (24.91) | 169 (24.78) | |
| High school or above | 8288 (80.12) | 7775 (80.46) | 513 (75.22) | | 2002 (75.12) | 1489 (75.09) | 513 (75.22) | |
| Gravidity, n (%) | | | | <0.05 | | | | 0.389 |
| <3 | 7439 (71.91) | 6989 (72.33) | 450 (65.98) | | 1794 (67.32) | 1344 (67.78) | 450 (65.98) | |
| >=3 | 2906 (28.09) | 2674 (27.67) | 232 (34.02) | | 871 (32.68) | 639 (32.22) | 232 (34.02) | |
| Parity, n (%) | | | | 0.170 | | | | 0.413 |
| Primipara | 6623 (64.02) | 6203 (64.19) | 420 (61.58) | | 1676 (62.89) | 1256 (63.34) | 420 (61.58) | |
| Multipara | 3722 (35.98) | 3460 (35.81) | 262 (38.42) | | 989 (37.11) | 727 (36.66) | 262 (38.42) | |
| Neonatal Sex, n (%) | | | | 0.142 | | | | 0.657 |
| Male | 5392 (52.12) | 5018 (51.93) | 374 (54.84) | | 1442 (54.11) | 1068 (53.86) | 374 (54.84) | |
| Female | 4953 (47.88) | 4645 (48.07) | 308 (45.16) | | 1223 (45.89) | 915 (46.14) | 308 (45.16) | |
| Floating Population, n (%) | | | | <0.05 | | | | 0.239 |
| Yes | 2804 (27.1) | 2578 (26.68) | 226 (33.14) | | 835 (31.33) | 609 (30.71) | 226 (33.14) | |
| No | 7541 (72.9) | 7085 (73.32) | 456 (66.86) | | 1830 (68.67) | 1374 (69.29) | 456 (66.86) | |
| Pregnancy Complication, n (%) | | | | <0.05 | | | | 0.786 |
| Yes | 3813 (36.86) | 3647 (37.74) | 166 (24.34) | | 659 (24.73) | 493 (24.86) | 166 (24.34) | |
| No | 6532 (63.14) | 6016 (62.26) | 516 (75.66) | | 2006 (75.27) | 1490 (75.14) | 516 (75.66) | |
| Fetal Position, n (%) | | | | <0.05 | | | | 0.162 |
| Cephalic | 9832 (95.04) | 9258 (95.81) | 574 (84.16) | | 2286 (85.78) | 1712 (86.33) | 574 (84.16) | |
| Non-cephalic | 513 (4.96) | 405 (4.19) | 108 (15.84) | | 379 (14.22) | 271 (13.67) | 108 (15.84) | |
| ART, n (%) | | | | <0.05 | | | | 0.326 |
| Yes | 776 (7.50) | 666 (6.89) | 110 (16.13) | | 399 (14.97) | 289 (14.57) | 110 (16.13) | |
| No | 9569 (92.50) | 8997 (93.11) | 572 (83.87) | | 2266 (85.03) | 1694 (85.43) | 572 (83.87) | |
| Economic Level, n (%) | | | | <0.05 | | | | 0.980 |
| Average and below | 8565 (82.79) | 7944 (82.21) | 621 (91.06) | | 2426 (91.03) | 1805 (91.02) | 621 (91.06) | |
| Above average | 1780 (17.21) | 1719 (17.79) | 61 (8.94) | | 239 (8.97) | 178 (8.98) | 61 (8.94) | |

PSM: propensity score matching; SD: standard deviations; CffDNA: cell-free fetal DNA; PTB: preterm birth; ART: Assisted reproductive technology.

all baseline characteristics achieved statistical balance (all p > 0.05) between matched groups – 682 PTB cases matched with 1,983 term controls and 533 LBW cases matched with 1,579 normal birth weight controls. Fig 2 illustrate the data distribution and standardized mean differences (SMDs) values before and after matching.

### Association between CffDNA and adverse neonatal outcomes: Preterm birth and low birth weight

**Univariate and multivariate logistic regression analysis of CffDNA and PTB/LBW.** In the preterm birth analysis (Table 3), pre-matching univariate analysis revealed a significant inverse association between cffDNA levels and preterm

**Table 2. Baseline Characteristics Before and After PSM for the Association Between CffDNA and LBW.**

| Variable | Before PSM | | | | After PSM | | | |
|---|---|---|---|---|---|---|---|---|
| | Total (n = 10345) | Non-LBW (n = 9807) | LBW (n = 538) | P | Total (n = 2112) | Non-LBW (n = 1579) | LBW (n = 533) | P |
| Age, n (%) | | | | <0.05 | | | | 0.591 |
| <35 | 8550 (82.65) | 8136 (82.96) | 414 (76.95) | | 1654 (78.31) | 1241 (78.59) | 413 (77.49) | |
| >=35 | 1795 (17.35) | 1671 (17.04) | 124 (23.05) | | 458 (21.69) | 338 (21.41) | 120 (22.51) | |
| BMI, n (%) | | | | 0.967 | | | | 0.734 |
| <30 | 8184 (79.11) | 7758 (79.11) | 426 (79.18) | | 1683 (79.69) | 1261 (79.86) | 422 (79.17) | |
| >=30 | 2161 (20.89) | 2049 (20.89) | 112 (20.82) | | 429 (20.31) | 318 (20.14) | 111 (20.83) | |
| Level of Education, n (%) | | | | <0.05 | | | | 0.643 |
| Less than high school | 2057 (19.88) | 1924 (19.62) | 133 (24.72) | | 523 (24.76) | 395 (25.02) | 128 (24.02) | |
| High school or above | 8288 (80.12) | 7883 (80.38) | 405 (75.28) | | 1589 (75.24) | 1184 (74.98) | 405 (75.98) | |
| Gravidity, n (%) | | | | <0.05 | | | | 0.758 |
| <3 | 7439 (71.91) | 7075 (72.14) | 364 (67.66) | | 1423 (67.38) | 1061 (67.19) | 362 (67.92) | |
| >=3 | 2906 (28.09) | 2732 (27.86) | 174 (32.34) | | 689 (32.62) | 518 (32.81) | 171 (32.08) | |
| Parity, n (%) | | | | 0.813 | | | | 0.542 |
| Primipara | 6623 (64.02) | 6276 (64.00) | 347 (64.50) | | 1386 (65.62) | 1042 (65.99) | 344 (64.54) | |
| Multipara | 3722 (35.98) | 3531 (36.00) | 191 (35.50) | | 726 (34.38) | 537 (34.01) | 189 (35.46) | |
| Neonatal Sex, n (%) | | | | 0.356 | | | | 0.982 |
| Male | 5392 (52.12) | 5122 (52.23) | 270 (50.19) | | 1065 (50.43) | 796 (50.41) | 269 (50.47) | |
| Female | 4953 (47.88) | 4685 (47.77) | 268 (49.81) | | 1047 (49.57) | 783 (49.59) | 264 (49.53) | |
| Floating Population, n (%) | | | | <0.05 | | | | 0.382 |
| Yes | 2804 (27.1) | 2633 (26.85) | 171 (31.78) | | 634 (30.02) | 466 (29.51) | 168 (31.52) | |
| No | 7541 (72.9) | 7174 (73.15) | 367 (68.22) | | 1478 (69.98) | 1113 (70.49) | 365 (68.48) | |
| Pregnancy Complication, n (%) | | | | <0.05 | | | | 0.879 |
| Yes | 3813 (36.86) | 3663 (37.35) | 150 (27.88) | | 585 (27.7) | 436 (27.61) | 149 (27.95) | |
| No | 6532 (63.14) | 6144 (62.65) | 388 (72.12) | | 1527 (72.3) | 1143 (72.39) | 384 (72.05) | |
| Fetal Position, n (%) | | | | <0.05 | | | | 0.373 |
| Cephalic | 9832 (95.04) | 9383 (95.68) | 449 (83.46) | | 1804 (85.42) | 1355 (85.81) | 449 (84.24) | |
| Non-cephalic | 513 (4.96) | 424 (4.32) | 89 (16.54) | | 308 (14.58) | 224 (14.19) | 84 (15.76) | |
| ART, n (%) | | | | <0.05 | | | | 0.662 |
| Yes | 776 (7.50) | 702 (7.16) | 74 (13.75) | | 262 (12.41) | 193 (12.22) | 69 (12.95) | |
| No | 9569 (92.50) | 9105 (92.84) | 464 (86.25) | | 1850 (87.59) | 1386 (87.78) | 464 (87.05) | |
| Economic Level, n (%) | | | | <0.05 | | | | 0.979 |
| Average and below | 8565 (82.79) | 8066 (82.25) | 499 (92.75) | | 1958 (92.71) | 1464 (92.72) | 494 (92.68) | |
| Above average | 1780 (17.21) | 1741 (17.75) | 39 (7.25) | | 154 (7.29) | 115 (7.28) | 39 (7.32) | |

PSM: propensity score matching; SD: standard deviations; CffDNA: cell-free fetal DNA; LBW: low birth weight; ART: Assisted reproductive technology.

birth risk across all quartiles (all P < 0.05). After adjusting for confounders, these inverse associations persisted in the pre-matching multivariate analysis (all P < 0.05). Post-matching analyses confirmed these findings, with univariate results showing maintained protection (all P < 0.05) and multivariate analysis demonstrating consistent effect sizes (all P < 0.05) Similarly, for low birth weight outcomes (Table 4), a more nuanced pattern emerged. In pre-matching analyses, only the highest cffDNA quartile showed significant inverse associations in both univariate (P < 0.05) and multivariate models (P < 0.05), while lower quartiles demonstrated non-significant associations. Post-matching analyses revealed similar

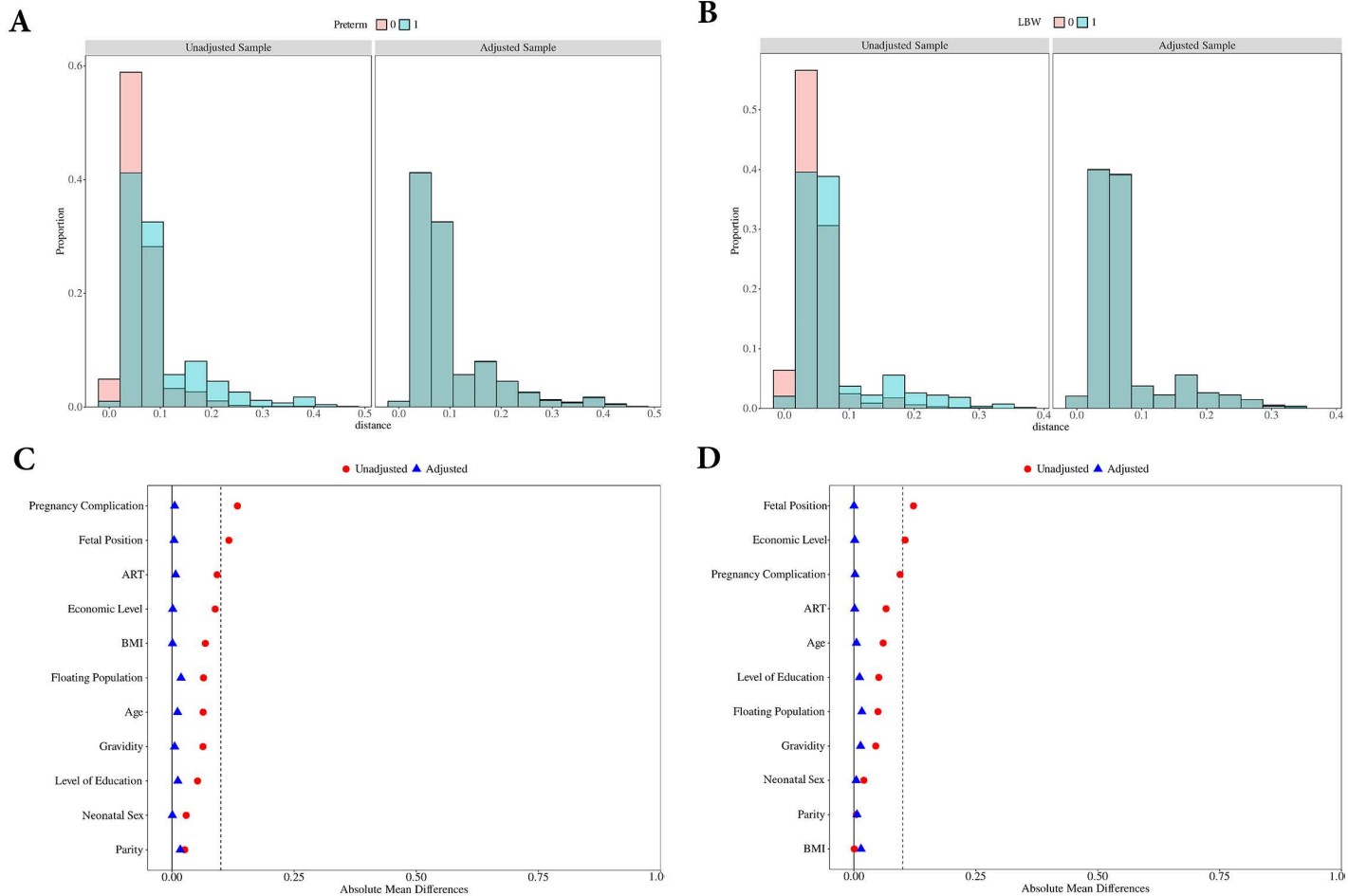

**Fig 2. Improvement of covariate balance after propensity score matching (PSM) in preterm birth and low birth weight cohorts. (A)** Data distribution for the preterm birth cohort in the unadjusted and PSM-adjusted samples. **(B)** Data distribution for the low birth weight cohort in the unadjusted and PSM-adjusted samples. **(C)** Standardized Mean Differences (SMDs) for covariates between groups before and after PSM for the preterm birth analysis. The results provided an intuitive visualization of balance improvement across all variables. **(D)** SMDs for covariates between groups before and after PSM for the low birth weight analysis. The results provided an intuitive visualization of balance improvement across all variables.

findings, with the highest quartile maintaining significant protection in both univariate (P<0.05) and multivariate models (P<0.05), whereas Q2 and Q3 quartiles continued to show non-significant associations with LBW risk.

**Hierarchical logistic regression models on the association of CffDNA and LBW/PTB.** Hierarchical logistic regression models were employed to systematically evaluate the association between cffDNA levels and risks of PTB and LBW through three progressively adjusted models (Fig 3): Model 1 (unadjusted), Model 2 (adjusted for basic covariates: age, BMI, level of education, gravidity, and parity), and Model 3 (fully adjusted for all relevant confounders including neonatal sex, floating population, pregnancy complication, fetal position, ART, and economic level). Analyses were conducted both before and after PSM to ensure robustness.For PTB outcomes, a consistent and significant inverse association was observed across all cffDNA quartiles in every model, both before and after PSM. In Model 1 (unadjusted) before PSM, higher cffDNA levels were associated with reduced PTB risk (Q2: OR=0.78, 95%CI:0.64–0.96; Q3: OR=0.67, 95%CI:0.54–0.83; Q4: OR=0.62, 95%CI:0.50–0.77; all P<0.05), a trend that persisted after PSM (Q2: OR=0.78,

**Table 3. Univariate and Multivariate Logistic Regression Analysis of CffDNA and PTB.**

| Variable | Before PSM | | | | After PSM | | | |
|---|---|---|---|---|---|---|---|---|
| | Univariate logistic | | Multivariable logistic | | Univariate logistic | | Multivariable logistic | |
| | OR (95%CI) | P | OR (95%CI) | P | OR (95%CI) | P | OR (95%CI) | P |
| Age | | | | | | | | |
| <35 | 1.00 (Reference) | | 1.00 (Reference) | | 1.00 (Reference) | | 1.00 (Reference) | |
| >=35 | 1.49 (1.24~1.80) | <0.05 | 1.25 (1.01~1.54) | <0.05 | 1.13 (0.92~1.39) | 0.244 | 1.09 (0.86~1.38) | 0.486 |
| BMI | | | | | | | | |
| <30 | 1.00 (Reference) | | 1.00 (Reference) | | 1.00 (Reference) | | 1.00 (Reference) | |
| >=30 | 1.46 (1.22~1.74) | <0.05 | 1.22 (1.02~1.47) | <0.05 | 1.04 (0.86~1.27) | 0.685 | 0.99 (0.81~1.21) | 0.924 |
| Level of Education | | | | | | | | |
| Less than high school | 1.00 (Reference) | | 1.00 (Reference) | | 1.00 (Reference) | | 1.00 (Reference) | |
| High school or above | 0.74 (0.62~0.88) | <0.05 | 0.98 (0.81~1.19) | 0.845 | 1.01 (0.82~1.23) | 0.945 | 1.04 (0.84~1.28) | 0.726 |
| Gravidity | | | | | | | | |
| <3 | 1.00 (Reference) | | 1.00 (Reference) | | 1.00 (Reference) | | 1.00 (Reference) | |
| >=3 | 1.35 (1.14~1.59) | <0.05 | 1.18 (0.95~1.48) | 0.139 | 1.08 (0.90~1.30) | 0.389 | 1.02 (0.80~1.30) | 0.896 |
| Parity | | | | | | | | |
| Primipara | 1.00 (Reference) | | 1.00 (Reference) | | 1.00 (Reference) | | 1.00 (Reference) | |
| Multipara | 1.12 (0.95~1.31) | 0.170 | 0.86 (0.70~1.07) | 0.191 | 1.08 (0.90~1.29) | 0.413 | 1.03 (0.81~1.31) | 0.820 |
| Neonatal Sex | | | | | | | | |
| Male | 1.00 (Reference) | | 1.00 (Reference) | | 1.00 (Reference) | | 1.00 (Reference) | |
| Female | 0.89 (0.76~1.04) | 0.142 | 0.88 (0.75~1.03) | 0.110 | 0.96 (0.81~1.14) | 0.657 | 0.95 (0.79~1.13) | 0.552 |
| Floating Population | | | | | | | | |
| Yes | 1.00 (Reference) | | 1.00 (Reference) | | 1.00 (Reference) | | 1.00 (Reference) | |
| No | 0.73 (0.62~0.87) | <0.05 | 0.84 (0.70~1.01) | 0.058 | 0.89 (0.74~1.08) | 0.239 | 0.90 (0.74~1.10) | 0.322 |
| Pregnancy Complication | | | | | | | | |
| No | 1.00 (Reference) | | 1.00 (Reference) | | 1.00 (Reference) | | 1.00 (Reference) | |
| Yes | 0.53 (0.44~0.64) | <0.05 | 0.62 (0.51~0.75) | <0.05 | 0.97 (0.79~1.19) | 0.786 | 1.01 (0.82~1.26) | 0.913 |
| Fetal Position | | | | | | | | |
| Cephalic | 1.00 (Reference) | | 1.00 (Reference) | | 1.00 (Reference) | | 1.00 (Reference) | |
| Non-cephalic | 4.30 (3.42~5.40) | <0.05 | 3.31 (2.60~4.22) | <0.05 | 1.19 (0.93~1.51) | 0.162 | 1.17 (0.91~1.51) | 0.209 |
| ART | | | | | | | | |
| No | 1.00 (Reference) | | 1.00 (Reference) | | 1.00 (Reference) | | 1.00 (Reference) | |
| Yes | 2.60 (2.09~3.23) | <0.05 | 2.13 (1.69~2.68) | <0.05 | 1.13 (0.89~1.43) | 0.326 | 1.10 (0.86~1.40) | 0.457 |
| Economic Level | | | | | | | | |
| Average and below | 1.00 (Reference) | | 1.00 (Reference) | | 1.00 (Reference) | | 1.00 (Reference) | |
| Above average | 0.45 (0.35~0.59) | <0.05 | 0.49 (0.37~0.65) | <0.05 | 1.00 (0.73~1.35) | 0.980 | 1.04 (0.76~1.42) | 0.818 |
| CffDNA | | | | | | | | |
| Q1 | 1.00 (Reference) | | 1.00 (Reference) | | 1.00 (Reference) | | 1.00 (Reference) | |
| Q2 | 0.78 (0.64~0.96) | <0.05 | 0.80 (0.65~0.99) | <0.05 | 0.78 (0.62~0.98) | <0.05 | 0.77 (0.61~0.97) | <0.05 |
| Q3 | 0.67 (0.54~0.83) | <0.05 | 0.69 (0.55~0.86) | <0.05 | 0.71 (0.56~0.91) | <0.05 | 0.71 (0.56~0.90) | <0.05 |
| Q4 | 0.62 (0.50~0.77) | <0.05 | 0.68 (0.54~0.85) | <0.05 | 0.69 (0.54~0.88) | <0.05 | 0.69 (0.54~0.88) | <0.05 |

CI: confidence intervals; OR:odds ratios; Quartiles: Q1 (0–25%), Q2 (25–50%), Q3 (50–75%), Q4 (75–100%); PSM: propensity score matching; CffDNA: cell-free fetal DNA; PTB: preterm birth.

**Table 4. Univariate and Multivariate Logistic Regression Analysis of CffDNA and LBW.**

| Variable | Before PSM | | | | After PSM | | | |
|---|---|---|---|---|---|---|---|---|
| | Univariate logistic | | Multivariable logistic | | Univariate logistic | | Multivariable logistic | |
| | OR (95%CI) | P | OR (95%CI) | P | OR (95%CI) | P | OR (95%CI) | P |
| Age | | | | | | | | |
| <35 | 1.00 (Reference) | | 1.00 (Reference) | | 1.00 (Reference) | | 1.00 (Reference) | |
| >=35 | 1.51 (1.19~1.93) | <0.05 | 1.31 (1.00~1.72) | 0.052 | 1.12 (0.87~1.42) | 0.379 | 1.05 (0.79~1.38) | 0.749 |
| BMI | | | | | | | | |
| <30 | 1.00 (Reference) | | 1.00 (Reference) | | 1.00 (Reference) | | 1.00 (Reference) | |
| >=30 | 1.02 (0.79~1.31) | 0.881 | 0.85 (0.66~1.11) | 0.231 | 0.77 (0.61~0.98) | <0.05 | 0.69 (0.54~0.89) | <0.05 |
| Level of Education | | | | | | | | |
| Less than high school | 1.00 (Reference) | | 1.00 (Reference) | | 1.00 (Reference) | | 1.00 (Reference) | |
| High school or above | 0.75 (0.59~0.96) | <0.05 | 0.94 (0.73~1.22) | 0.654 | 1.00 (0.79~1.27) | 0.996 | 1.01 (0.79~1.30) | 0.909 |
| Gravidity | | | | | | | | |
| <3 | 1.00 (Reference) | | 1.00 (Reference) | | 1.00 (Reference) | | 1.00 (Reference) | |
| >=3 | 1.30 (1.04~1.62) | <0.05 | 1.19 (0.89~1.59) | 0.246 | 1.03 (0.83~1.28) | 0.778 | 0.95 (0.71~1.26) | 0.704 |
| Parity | | | | | | | | |
| Primipara | 1.00 (Reference) | | 1.00 (Reference) | | 1.00 (Reference) | | 1.00 (Reference) | |
| Multipara | 1.08 (0.87~1.33) | 0.498 | 0.85 (0.64~1.13) | 0.272 | 1.04 (0.84~1.28) | 0.749 | 0.99 (0.75~1.32) | 0.948 |
| Neonatal Sex | | | | | | | | |
| Male | 1.00 (Reference) | | 1.00 (Reference) | | 1.00 (Reference) | | 1.00 (Reference) | |
| Female | 1.04 (0.85~1.28) | 0.707 | 1.03 (0.84~1.27) | 0.757 | 1.13 (0.92~1.39) | 0.244 | 1.12 (0.91~1.38) | 0.283 |
| Floating Population | | | | | | | | |
| Yes | 1.00 (Reference) | | 1.00 (Reference) | | 1.00 (Reference) | | 1.00 (Reference) | |
| No | 0.87 (0.69~1.09) | 0.212 | 0.96 (0.75~1.22) | 0.737 | 0.88 (0.71~1.09) | 0.238 | 0.89 (0.70~1.12) | 0.307 |
| Pregnancy Complication | | | | | | | | |
| No | 1.00 (Reference) | | 1.00 (Reference) | | 1.00 (Reference) | | 1.00 (Reference) | |
| Yes | 0.63 (0.50~0.80) | <0.05 | 0.75 (0.59~0.95) | <0.05 | 1.10 (0.87~1.39) | 0.419 | 1.20 (0.93~1.54) | 0.154 |
| Fetal Position | | | | | | | | |
| Cephalic | 1.00 (Reference) | | 1.00 (Reference) | | 1.00 (Reference) | | 1.00 (Reference) | |
| Non-cephalic | 4.17 (3.10~5.60) | <0.05 | 3.42 (2.50~4.66) | <0.05 | 1.65 (1.27~2.15) | <0.05 | 1.75 (1.32~2.30) | <0.05 |
| ART | | | | | | | | |
| No | 1.00 (Reference) | | 1.00 (Reference) | | 1.00 (Reference) | | 1.00 (Reference) | |
| Yes | 2.05 (1.52~2.77) | <0.05 | 1.71 (1.25~2.34) | <0.05 | 1.04 (0.78~1.38) | 0.790 | 1.07 (0.80~1.43) | 0.660 |
| Economic Level | | | | | | | | |
| Average and below | 1.00 (Reference) | | 1.00 (Reference) | | 1.00 (Reference) | | 1.00 (Reference) | |
| Above average | 0.42 (0.29~0.61) | <0.05 | 0.44 (0.30~0.63) | <0.05 | 0.48 (0.31~0.76) | <0.05 | 0.48 (0.30~0.76) | <0.05 |
| CffDNA | | | | | | | | |
| Q1 | 1.00 (Reference) | | 1.00 (Reference) | | 1.00 (Reference) | | 1.00 (Reference) | |
| Q2 | 0.98 (0.75~1.29) | 0.887 | 0.97 (0.74~1.28) | 0.826 | 0.86 (0.66~1.13) | 0.284 | 0.85 (0.65~1.12) | 0.261 |
| Q3 | 0.87 (0.65~1.15) | 0.315 | 0.86 (0.65~1.15) | 0.318 | 0.83 (0.63~1.10) | 0.206 | 0.81 (0.61~1.08) | 0.144 |
| Q4 | 0.65 (0.48~0.88) | <0.05 | 0.67 (0.49~0.91) | <0.05 | 0.69 (0.52~0.93) | <0.05 | 0.67 (0.50~0.90) | <0.05 |

CI: confidence intervals; OR:odds ratios; Quartiles: Q1 (0–25%), Q2 (25–50%), Q3 (50–75%), Q4 (75–100%); PSM: propensity score matching; CffDNA: cell-free fetal DNA; LBW: low birth weight.

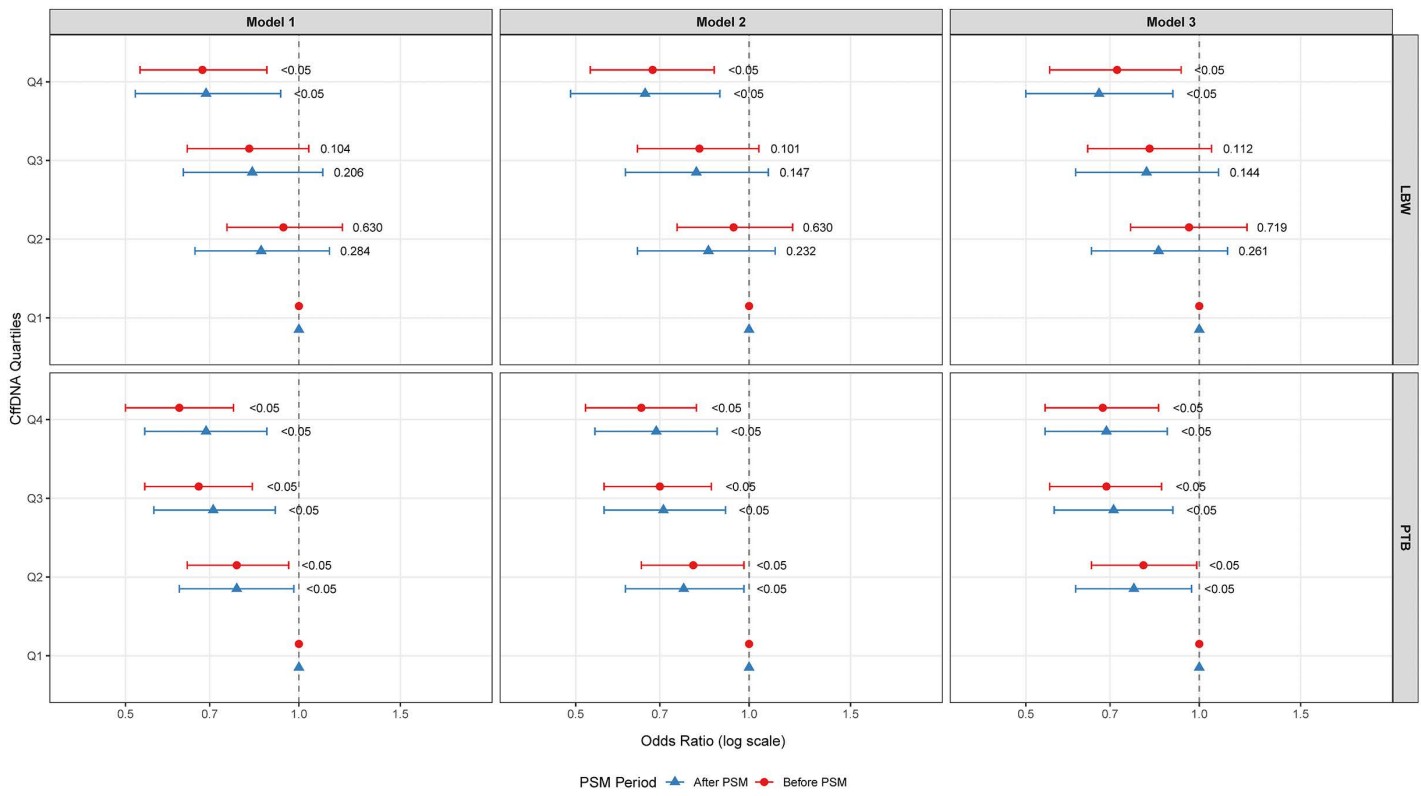

**Fig 3. Association between cffDNA levels and risks of LBW and PTB from hierarchical logistic regression models.** Three sequentially adjusted models are shown: Model 1 (crude association); Model 2 (adjusted for maternal age, BMI, education, gravidity, and parity); and Model 3 (further adjusted for neonatal sex, floating population status, pregnancy complications, fetal position, ART, and economic level).

95%CI:0.62–0.98; Q3: OR=0.71, 95%CI:0.56–0.91; Q4: OR=0.69, 95%CI:0.54–0.88; all P < 0.05). In Model 2 (partially adjusted), the inverse associations remained significant before PSM (Q2: aOR=0.80, 95%CI:0.65–0.98; Q3: aOR=0.70, 95%CI:0.56–0.86; Q4: aOR=0.65, 95%CI:0.52–0.81; all P < 0.05) and after PSM (Q2: aOR=0.77, 95%CI:0.61–0.98; Q3: aOR=0.71, 95%CI:0.56–0.91; Q4: aOR=0.69, 95%CI:0.54–0.88; all P < 0.05). Finally, in the fully adjusted Model 3, the associations were robustly maintained before PSM (Q2: aOR=0.80, 95%CI:0.65–0.99; Q3: aOR=0.69, 95%CI:0.55–0.86; Q4: aOR=0.68, 95%CI:0.54–0.85; all P < 0.05) and after PSM (Q2: aOR=0.77, 95%CI:0.61–0.97; Q3: aOR=0.71, 95%CI:0.56–0.90; Q4: aOR=0.69, 95%CI:0.54–0.88; all P < 0.05).For LBW outcomes, in Model 1 before PSM, only the highest quartile (Q4) showed a significant inverse association (Q4: OR=0.68, 95%CI:0.53–0.88; P < 0.05), which was confirmed after PSM (Q4: OR=0.69, 95%CI:0.52–0.93; P < 0.05). In Model 2, this Q4-specific effect remained stable before PSM (aOR=0.68, 95%CI:0.53–0.87; P < 0.05) and after PSM (aOR=0.66, 95%CI:0.49–0.89; P < 0.05). Similarly, in the fully adjusted Model 3, the significant protection for Q4 was consistent before PSM (aOR=0.72, 95%CI:0.55–0.93; P < 0.05) and after PSM (aOR=0.67, 95%CI:0.50–0.90; P < 0.05), while associations for Q2 and Q3 remained non-significant across all models.

**Subgroup analyses before and after PSM.** Using a cutoff of cffDNA = 16 as a binary variable, we systematically evaluated the heterogeneity of its associations with PTB and LBW across subgroups with different demographic and clinical characteristics. In the pre-PSM analysis, (Fig 4A) cffDNA ≥ 16 was significantly associated with a reduced risk of PTB (overall OR = 0.70, 95% CI: 0.60–0.81, P < 0.001). This protective association remained consistent across most subgroups, including key categories such as age < 35 years (OR = 0.68, 95% CI: 0.57–0.82), BMI < 30

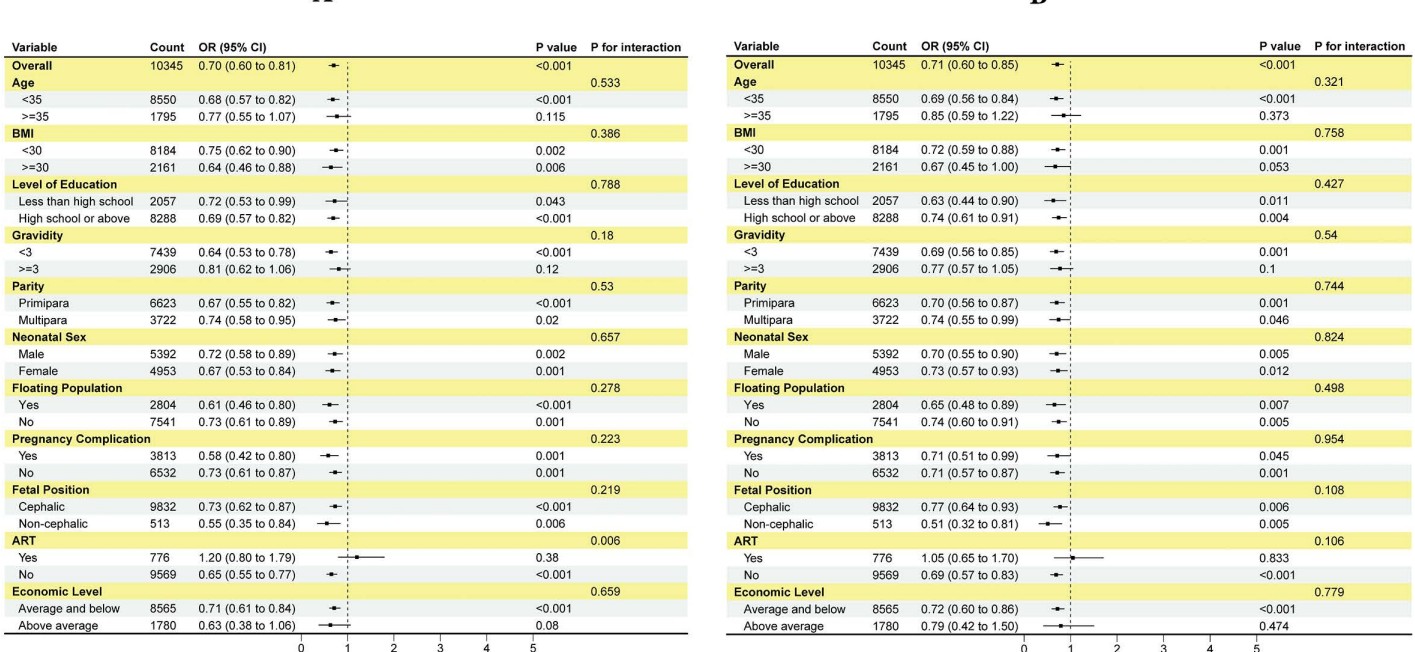

**Fig 4. Subgroup analysis of the relationship between cffDNA and adverse birth outcomes before PSM. (A)** Before PSM, subgroup analysis of the relationship between cffDNA and PTB. A cutoff value of 16 was used to convert cffDNA into a categorical variable. **(B)** Before PSM, subgroup analysis of the relationship between cffDNA and LBW. A cutoff value of 16 was used to convert cffDNA into a categorical variable.

(OR = 0.75, 95% CI: 0.62–0.90), primiparity (OR = 0.67, 95% CI: 0.55–0.82), and cephalic fetal position (OR = 0.73, 95% CI: 0.61–0.87). Interaction analysis indicated that, except for the assisted reproduction subgroup, all other subgroups showed P-interaction > 0.05, suggesting no significant effect modification. Similarly, (Fig 4B) cffDNA ≥ 16 was significantly associated with a lower risk of LBW (overall OR = 0.71, 95% CI: 0.60–0.85, P < 0.001). This association was also observed in subgroups such as age < 35 years (OR = 0.69, 95% CI: 0.56–0.84), BMI < 30 (OR = 0.72, 95% CI: 0.59–0.88), and primiparity (OR = 0.70, 95% CI: 0.56–0.87). All interaction tests for LBW were non-significant (P-interaction > 0.05), indicating a relatively consistent association pattern across different population characteristics.The post-PSM analysis further confirmed the robustness of these findings. For PTB, (Fig 5A) the inverse association of cffDNA ≥ 16 remained significant (overall OR = 0.78, 95% CI: 0.65–0.95, P = 0.006) and was particularly evident in subgroups such as BMI ≥ 30 (OR = 0.62, 95% CI: 0.43–0.85, P = 0.007) and multiparity (OR = 0.70, 95% CI: 0.53–0.93, P = 0.014). Notably, a significant interaction was observed in the ART subgroup (P-interaction < 0.001), where the inverse association of cffDNA ≥ 16 was more pronounced in the naturally conceived group (OR = 0.70, 95% CI: 0.58–0.86). For LBW, (Fig 5B) cffDNA ≥ 16 continued to show a protective trend after PSM (overall OR = 0.79, 95% CI: 0.65–0.96, P = 0.019), reaching statistical significance in subgroups such as age < 35 years (OR = 0.78, 95% CI: 0.62–0.97, P = 0.027) and male newborns (OR = 0.75, 95% CI: 0.57–0.99, P = 0.043). All interaction P-values for LBW remained > 0.05, further supporting the robustness of the association.

**Threshold effect analysisof cffDNA in relation to preterm birth and low birth weight risk.** A threshold effect analysis was conducted to evaluate the association between cffDNA and the risks of PTB and LBW before and after propensity score matching (PSM). Before PSM, for PTB, a significant threshold was identified at a cffDNA level of 17.59 (Fig 6A). Below this threshold, each unit increase in cffDNA was significantly associated with a reduced risk of PTB

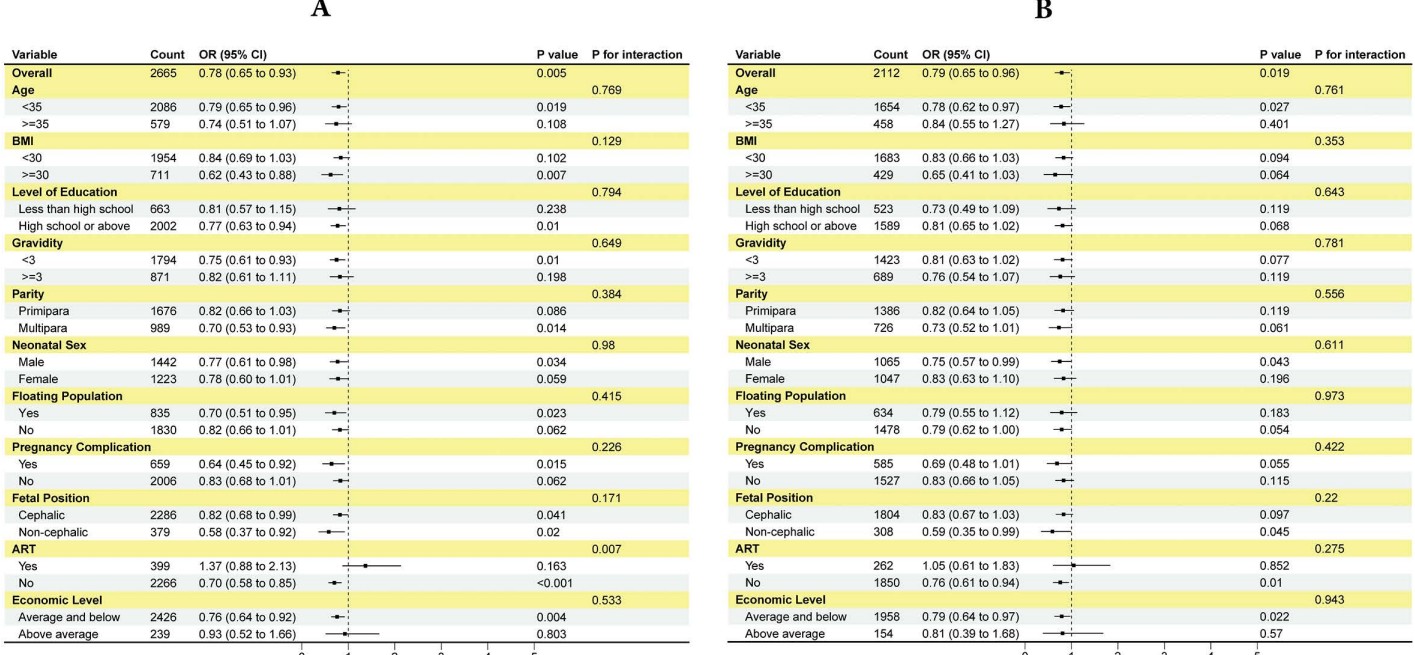

**Fig 5. Subgroup analysis of the relationship between cffDNA and adverse birth outcomes after PSM. (A)** After PSM, subgroup analysis of the relationship between cffDNA and PTB. A cutoff value of 16 was used to convert cffDNA into a categorical variable. **(B)** After PSM, subgroup analysis of the relationship between cffDNA and LBW. A cutoff value of 16 was used to convert cffDNA into a categorical variable.

(OR = 0.95, 95% CI: 0.92–0.98, P < 0.05), whereas above the threshold, no significant association was observed (OR = 0.99, 95% CI: 0.93–1.06, P = 0.838). For LBW, the threshold was identified at a cffDNA level of 22.61 (Fig 6B). Below this value, a significant inverse associationwas found (OR = 0.97, 95% CI: 0.95–0.99, P < 0.05), while above it, the association was not significant (OR = 0.40, 95% CI: 0.11–1.48, P = 0.170). After PSM, for PTB, the threshold shifted to a cffDNA level of 12.23 (Fig 6C). A significant protective association remained below this new threshold (OR = 0.90, 95% CI: 0.82–0.99, P < 0.05), but no significant association was observed above it (OR = 0.98, 95% CI: 0.95–1.01, P = 0.163). For LBW, the threshold was identified at 14.29 (Fig 6D); however, no significant association was found either below (OR = 1.00, 95% CI: 0.93–1.07, P = 0.975) or above (OR = 0.96, 95% CI: 0.92–1.00, P = 0.053) this cutoff after matching.

## Discussion

This study demonstrates a significant inverse association between elevated cffDNA levels and reduced risks of both preterm birth (PTB) and low birth weight (LBW), with notable differences in the nature of these relationships. For PTB, we observed a consistent inverse association across all cffDNA quartiles that remained robust across multiple analytical approaches—univariate and multivariate logistic regression, hierarchical modeling, and subgroup analyses—both before and after propensity score matching. In contrast, the association with LBW was more nuanced, with significant inverse associations observed only in the highest cffDNA quartile.

The consistent inverse association between cffDNA and PTB across all analytical approaches suggests a potentially robust biological relationship. At first glance, this finding may appear to contrast with traditional interpretations of cffD-NAas a marker of placental stress or injury, where elevated levels might be expected to correlate with adverse outcomes. However, this apparent paradox can be reconciled by distinguishing between two distinct biological contexts that lead to elevated cffDNA. On one hand, pathological elevations in cffDNA can occur as a result of placental damage, inflammation,

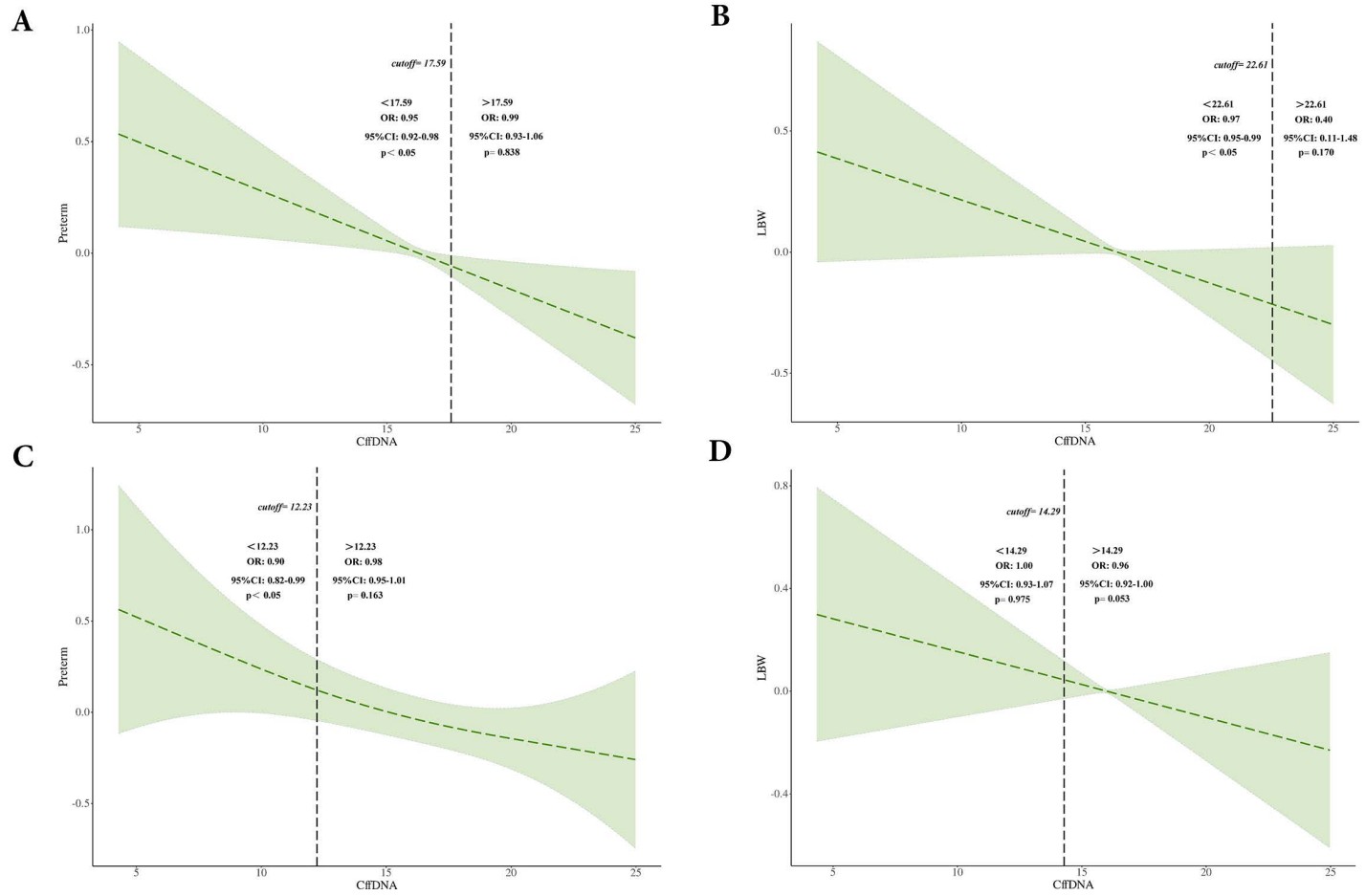

**Fig 6. Threshold effect analysis of cffDNA on preterm birth (PTB) and low birth weight (LBW) before and after PSM. (A)** Association between cffDNA and PTB in the unadjusted cohort. The optimal threshold was identified at 17.59%. **(B)** Association between cffDNA and PTB in the PSM-adjusted cohort. The optimal threshold was identified at 22.61%. **(C)** Association between cffDNA and LBW in the unadjusted cohort. The optimal threshold was identified at 12.23%. **(D)** Association between cffDNA and LBW in the PSM-adjusted cohort. The optimal threshold was identified at 14.29%.

or apoptosis secondary to hypoxic or oxidative stress—situations where the placenta is under duress and at increased risk of dysfunction. In such contexts, higher cffDNA may signal placental injury and predict adverse outcomes. On the other hand, physiologically higher cffDNA levels in the first trimester may simply reflect greater placental mass, more robust trophoblast invasion, and healthy, regulated trophoblast turnover—all indicators of optimal placental development and function. Our findings align with this latter interpretation: in a generally low-risk population without overt placental pathology, higher cffDNA likely represents better placentation rather than incipient injury. This distinction is critical: the same biomarker can have different implications depending on the clinical context, timing of measurement, and underlying placental biology. Higher cffDNA levels may reflect better placental development and function, as cffDNA primarily originates from apoptotic trophoblasts [16]. The placenta plays a crucial role in maintaining pregnancy, and impaired placental function has been extensively linked to preterm labor [17]. Our findings align with emerging research suggesting that the quantity and quality of cell-free nucleic acids in maternal circulation are biomarkers of placental health [18]. The persistence of this association after rigorous adjustment for confounders and propensity score matching strengthens the validity of

this relationship.The threshold effect observed for LBW, where only the highest cffDNA quartile demonstrated significant protection, suggests a different underlying mechanism. This pattern may indicate that a certain minimum level of trophoblast turnover is necessary to support optimal fetal growth, but beyond this threshold, additional increases in cffDNA do not provide further benefits. Alternatively, this could reflect non-linear relationships between placental function and fetal growth, where only substantially better placental function (as indicated by higher cffDNA) translates to measurable protection against LBW. This finding is consistent with studies suggesting that fetal growth restriction often results from more severe placental dysfunction, including impaired maternal vascular perfusion, than preterm birth [19].

The threshold effect analysis provided further insights into these relationships. The identification of significant thresholds for both PTB and LBW before PSM, with only the PTB threshold remaining significant after matching, suggests that the cffDNA-LBW relationship may be more susceptible to confounding factors. The shifting threshold values after PSM highlight the importance of accounting for baseline differences when evaluating continuous biomarker-disease relationships, a methodological consideration emphasized in recent pharmacoepidemiological studies [20]. Our subgroup analyses revealed generally consistent inverse associations of cffDNA (using the ≥ 16 cutoff) across most demographic and clinical subgroups for both PTB and LBW. The significant interaction observed in the ART subgroup for PTB is particularly noteworthy, suggesting that the relationship between cffDNA and pregnancy outcomes may differ in assisted versus naturally conceived pregnancies. This finding merits further investigation, as ART pregnancies are known to have altered placental DNA methylation patterns and different pregnancy outcomes [21]. The consistency of the association across most other subgroups enhances the generalizability of our core finding.Several mechanisms could explain the protective associations observed. Higher cffDNA levels may indicate adequate trophoblast invasion and placental maturation, which are essential for maintaining pregnancy and supporting fetal growth [22]. Alternatively, optimal cffDNA levels might reflect appropriate balance in apoptotic processes, whereas both insufficient and excessive apoptosis could contribute to placental dysfunction. The inflammatory response associated with placental dysfunction has been linked to both PTB and LBW [23], and cffDNA may serve as a marker of this underlying process. Recent studies have further elucidated that cell-free nucleic acids can actively participate in inflammatory pathways, potentially providing a mechanistic link to adverse pregnancy outcomes [24].

From a clinical perspective, an important question is whether these robust associations translate into meaningful improvements in risk prediction and individual pregnancy care. Becking et al. [25] recently addressed this question in a large nationwide cohort of 56,110 pregnancies, demonstrating that while fetal fraction shows statistically significant associations with adverse outcomes, its added prognostic value beyond established clinical risk factors is limited. In their study, adding fetal fraction to prediction models resulted in only modest improvements in discrimination, with increases in the area under the curve ranging from 0.01 to 0.02 and small but significant integrated discrimination improvement indices, leading the authors to conclude that fetal fraction has "statistically significant but limited prognostic value". These findings contextualize our own results: the protective associations we observed, while consistent and biologically plausible, may not be sufficient to use cffDNA as a standalone predictor in clinical practice. However, because fetal fraction is routinely measured as a quality parameter in non-invasive prenatal testing without additional cost, it could still contribute to broader risk stratification strategies when integrated with other clinical parameters. Future research should focus on developing and validating multi-marker models that incorporate cffDNA alongside established predictors, and on evaluating whether such approaches can meaningfully improve perinatal outcomes through targeted interventions.

Key strengths of our study include the large sample size encompassing over ten thousand participants, which provides substantial statistical power and enhances the generalizability of our findings. Furthermore, we employed multiple analytical approaches, rigorous control for confounding through both multivariate adjustment and propensity score matching, and comprehensive subgroup and threshold analyses. The consistency of findings across these different methodological approaches enhances the credibility and robustness of our conclusions.However, several limitations should be acknowledged. The observational nature of our study precludes definitive causal conclusions. While we

controlled for numerous potential confounders, residual confounding remains possible. The measurement of cffDNA at a single time point limits our ability to assess dynamic changes throughout pregnancy, as the trajectory of cffDNA levels may be more informative than a single measurement. Additionally, LBW is a composite outcome influenced by both gestational age at delivery and fetal growth restriction. Our analysis did not adjust for gestational age (e.g., using small for gestational age [SGA] as an outcome), which limits the clinical interpretation of the LBW findings, as the observed inverse associations may be primarily driven by the prevention of preterm birth rather than enhanced fetal growth. Future studies should incorporate gestational-age-corrected birthweight measures to disentangle these pathways. Furthermore,as described in the Methods section, NIPT in our institution was offered to women with specific indications, including intermediate-risk results on first-trimester serum screening, contraindications to invasive procedures, or voluntary requests for aneuploidy risk assessment. Therefore, our cohort represents a selected subgroup of the pregnant population rather than an unselected general obstetric cohort. This selection may introduce bias, as women undergoing NIPT based on these indications may have different baseline characteristics and risk profiles compared to the broader population of pregnant women. For instance, women with intermediate-risk serum screening results may have distinct placental or biochemical profiles that could influence both cffDNA levels and pregnancy outcomes. Consequently, the observed associations between cffDNA and PTB/LBW may not be directly generalizable to lower-risk populations, to settings with different NIPT utilization policies, or to populations where NIPT is offered as a first-tier screening test to all pregnant women. Future studies in unselected cohorts or diverse healthcare settings with varying NIPT access criteria are warranted to validate our findings and assess their external validity.Finally, the subgroup and threshold analyses presented in this study were exploratory and not pre-specified. The multiple comparisons performed increase the risk of type I errors, and the data-driven nature of threshold selection may limit the reproducibility of the identified cutoff values. Therefore, these findings should be considered hypothesis-generating, and future studies in independent cohorts are needed to validate the observed patterns and interactions.while our sample is large, the generalizability of our findings to other ethnic or geographic populations requires further verification.

## Conclusion

In conclusion, our study provides compelling evidence that higher cffDNA levels are associated with reduced risks of both PTB and LBW, though with different patterns of association. For PTB, the relationship appears graded across all cffDNA levels, while for LBW, significant protection is observed only at the highest cffDNA levels. These findings suggest that cffDNA may serve as a useful biomarker for identifying pregnancies at risk for adverse outcomes, potentially enabling targeted monitoring and interventions.Future research should focus on validating these findings in diverse populations, investigating the underlying biological mechanisms, and exploring the potential clinical utility of serial cffDNA measurement in prenatal risk assessment.

### Data curation statement

The dataset used in this study was managed and curated under strict protocols to ensure data integrity and security. Access to the original dataset was restricted to the corresponding author, who performed all data cleaning, validation, and preprocessing steps. All analytical procedures were conducted by First author, with independent verification performed by corresponding author to ensure reproducibility. Any data-related inquiries may be directed to the corresponding author.

### Supporting information

**S1 Dataset. Dataset-the complete dataset of this study.**
(CSV)

## Acknowledgments

We thank all participants and researchers involved in the study.

## Author contributions

**Conceptualization:** Xiaosa Wang, Hui Shao, Guofeng Yin.

**Data curation:** Xiaosa Wang, Hui Shao, Junjiang Wu, Guofeng Yin.

**Formal analysis:** Hui Shao, Guofeng Yin.

**Funding acquisition:** Guofeng Yin.

**Methodology:** Xiaosa Wang, Hui Shao, Junjiang Wu.

**Writing – original draft:** Xiaosa Wang, Hui Shao, Junjiang Wu.

**Writing – review & editing:** Xiaosa Wang, Hui Shao, Guofeng Yin.

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
