## [Editor Report · Decision Letter 0]

23 Nov 2025

PONE-D-25-60974Association Between First-Trimester Cell-Free Fetal DNA Levels and the Risk of Preterm Birth and Low Birth Weight: A Propensity Score-Matched Cohort StudyPLOS ONE

Dear Dr. Yin,

Thank you for submitting your manuscript to PLOS ONE. After careful consideration, we feel that it has merit but does not fully meet PLOS ONE’s publication criteria as it currently stands. Therefore, we invite you to submit a revised version of the manuscript that addresses the points raised during the review process.

We look forward to receiving your revised manuscript.

Kind regards,

Preenan Pillay

Academic Editor

PLOS ONE

Journal Requirements:

2. Please amend the manuscript submission data (via Edit Submission) to include author “Xiaosa Wang”.

3. Please amend your authorship list in your manuscript file to include author “Xiasa Wang”.

Additional Editor Comments:

The following must be corrected to prevent rejection of the manuscript:

1. Introduction: the rationale of the study makes preemptive assumptions which must be avoided.  The rationale should specify the aim of the study with the approach to solving the challenge with a risk based stratified approach. 2. The methods do not comply and must be corrected as follows:- The molecular methods used for cffDNA must be included.

- The is a statement made about date of accessing data.  Was data used from another study or source, if yes please specify.- Please specify the reason for the removal of informed consent as this goes against the declaration of helinski Please provide a valid justification.  Failure would result in rejection of the manuscript.- I suggest that a methodology map be created to explain the detailed statistical analysis done per a question that is trying to be answered. In the results section  more graphical presentation of data is required.

---

## [Author Response · Author response to Decision Letter 1]

1 Dec 2025

Dear editors and reviewers,

We deeply appreciate your valuable suggestions and comments, which enable us to further improve our paper. We have studied the comments carefully and have made our best efforts to revise the entire paper as suggested by the editor and reviewers. We are very pleased that our paper can be revised, some detailed responses are as follows.

Journal Requirements:

Response : Thank you for the reminder, we have updated the manuscript to comply with PLOS ONE's style requirements.

2. Please amend the manuscript submission data (via Edit Submission) to include author “Xiaosa Wang”.

Response : Thank you for your message. We have updated the submission data via the "Edit Submission" feature to include author Xiaosa Wang as requested.

3. Please amend your authorship list in your manuscript file to include author “Xiasa Wang”.

Response : Thank you for your guidance. We have amended the authorship list in our manuscript file to include Xiasa Wang and have uploaded the revised version to the submission system.

Response : Thank you for the note. We confirm that no additional citation requests were made by the reviewers.

To Additional Editor:

1. Introduction: the rationale of the study makes preemptive assumptions which must be avoided. The rationale should specify the aim of the study with the approach to solving the challenge with a risk based stratified approach.

Response : We sincerely thank you for the valuable comment regarding the rationale of our study. In response to the concern about preemptive assumptions in the introduction, we have thoroughly revised the third and fourth paragraphs of the Introduction section. The modifications include reframing the study rationale around the existing uncertainty regarding first-trimester cell-free fetal DNA (cffDNA) as a biomarker for risk stratification, clearly stating the study aim to address this gap, and explicitly describing the use of cffDNA quartiles and multiple adjusted regression models as part of a risk-based analytical approach. All changes have been highlighted in red in the manuscript text for easy reference.

2. The methods do not comply and must be corrected as follows:

- The molecular methods used for cffDNA must be included.

- The is a statement made about date of accessing data. Was data used from another study or source, if yes please specify.

- Please specify the reason for the removal of informed consent as this goes against the declaration of helinski Please provide a valid justification. Failure would result in rejection of the manuscript.

- I suggest that a methodology map be created to explain the detailed statistical analysis done per a question that is trying to be answered.

Response : (1) We thank you for raising this important point. In response to the comment regarding molecular methods for cffDNA quantification, we have now added a detailed description in the Data Collection subsection of the Methods. Specifically, we have included the following information marked in red: "Cell-free fetal DNA (cffDNA) levels, expressed as fetal fraction (%), were quantitatively measured through clinical non-invasive prenatal testing (NIPT). The analysis was performed following standard protocols: maternal peripheral blood samples were collected, maternal plasma separation, DNA extraction, next-generation sequencing on the Illumina platform, and bioinformatic quantification of the fetal DNA fraction." This addition provides the technical details of cffDNA measurement.

(2) We thank you for this important question regarding data access. The data used in this study were original clinical data retrospectively collected from the electronic medical records of Shaoxing Maternal and Child Health Care Hospital for the specific purpose of this research. They are not sourced from an external database or a previously published study. The date “1 May 2025” mentioned in the manuscript was intended to indicate the point at which our final study dataset was locked for analysis, not the date of accessing an external source.

(3) We sincerely thank you for raising this critical ethical issue. The waiver of informed consent for this retrospective study was formally approved by our Institutional Review Board (Approval No: [IRB-AF-023-01.5]) as it aligns with international ethical guidelines under the following justifications: 1. the research involves no more than minimal risk as it exclusively utilizes fully de-identified clinical data; 2. obtaining individual consent is impracticable given the large-scale historical cohort (n=10,345) where participant re-contact is impossible; 3. the waiver does not adversely affect the rights or welfare of participants; 4. robust privacy protection measures were implemented, including permanent deletion of all direct identifiers and secure data encryption. For your convenience, we have uploaded the official ethical approval documents (both Chinese and English versions) which explicitly state the waiver of informed consent, to the submission system. We are confident that our approach fully complies with all regulatory requirements.

(4) We thank you for this valuable suggestion. In response, we have created a comprehensive methodology flowchart that visually outlines the entire analytical plan, from the initial cohort selection and propensity score matching to the specific statistical analysis applied to each research question. This figure has now been included as Figure 1 in the main manuscript. We believe it greatly enhances the clarity of our statistical approach.

3. In the results section more graphical presentation of data is required.

Response : We sincerely thank you for the suggestion to enhance the graphical presentation of our results. In direct response to this comment, we have now created additional figures to visually summarize the key findings of our study, bringing the total to six figures. Regarding the more detailed results in Tables 1-4, which present the baseline characteristics and the comprehensive univariate/multivariable logistic regression outputs before and after propensity score matching, we have deliberately kept these in a tabular format. We believe that for these specific sets of data, tables remain the clearest and most conventional method to display a large number of variables and statistical parameters, ensuring precision and facilitating easy comparison for the reader. Converting these extensive tables into figures would risk creating visual complexity that could obscure the detailed findings. We hope that the current structure of six figures and four tables provides an optimal balance, offering both high-level visual insight and the necessary granularity for in-depth scientific scrutiny. We hope the revised manuscript now meets with your approval.

We are very grateful for your insightful comments and suggestions, which are extremely valuable and helpful in improving the quality of our manuscript.

Sincerely yours,

Xiaosa Wang, Hui Shao, Junjiang Wu, Guofeng Yin

---

## [Decision Letter · Decision Letter 1]

15 Feb 2026

PONE-D-25-60974R1Association Between First-Trimester Cell-Free Fetal DNA Levels and the Risk of Preterm Birth and Low Birth Weight: A Propensity Score-Matched Cohort StudyPLOS One

Dear Dr. Yin,

Thank you for submitting your manuscript to PLOS ONE. After careful consideration, we feel that it has merit but does not fully meet PLOS ONE’s publication criteria as it currently stands. Therefore, we invite you to submit a revised version of the manuscript that addresses the points raised during the review process.

For the manuscript to be accepted: **- There are a number of technical elements that requires revision - kindly address all the reviewers' comments and provide a rebuttal document which clearly outlines the changes made in the manuscript for final consideration.****- Ensure that all grammar and English errors are appropriately corrected**

We look forward to receiving your revised manuscript.

Kind regards,

Preenan Pillay

Academic Editor

PLOS One

**Journal Requirements:**

**Additional Editor Comments:**

For the manuscript to be accepted all reviewers' comments must be addressed with a clear rebuttal letter that tracks all changes in the manuscript.

Reviewers' comments:

Reviewer's Responses to Questions

**Comments to the Author**

1. If the authors have adequately addressed your comments raised in a previous round of review and you feel that this manuscript is now acceptable for publication, you may indicate that here to bypass the “Comments to the Author” section, enter your conflict of interest statement in the “Confidential to Editor” section, and submit your "Accept" recommendation.

Reviewer #1: (No Response)

Reviewer #2: (No Response)

2. Is the manuscript technically sound, and do the data support the conclusions?

Reviewer #1: Yes

Reviewer #2: Yes

3. Has the statistical analysis been performed appropriately and rigorously? 

Reviewer #1: Yes

Reviewer #2: Yes

4. Have the authors made all data underlying the findings in their manuscript fully available?

Reviewer #1: Yes

Reviewer #2: Yes

5. Is the manuscript presented in an intelligible fashion and written in standard English?

Reviewer #1: Yes

Reviewer #2: Yes

6. Review Comments to the Author

Reviewer #1: This revised manuscript to me is scientifically sound. It shows the association between first-trimester fetal fraction and preterm birth and low birth weight. It adds to the body of work on this subject. I have some minor issues that I believe should be addressed.

- In the Objective in the Abstract add 'first trimester'

- For readability, I suggest not to put all the OR's/aOR's and 95%-CI's in the text of the Result section. They are already presented in the tables.

- In the Discussion section, could the authors provide more insight from a clinical point of view how fetal fraction could be a useful biomarker? Indeed, associations with adverse pregnancy outcomes have been shown, but does it really help to predict adverse outcomes and change individual pregnancy care? Possibly comment hereon also in light of a publication by Becking et al. on the predictive value of fetal fraction (Becking et al. BJOG, 2024; https://doi.org/10.1111/1471-0528.17978)

- Last, LWB without correcting for gestational age has little clinical meaning, this should be mentioned as a limitation of the study. Or are gestational-age corrected birthweights available?

Reviewer #2: FINDING ON THE AUTHOR'S RESPONSE TO THE 1ST REVIEWER'S CONCERNS.

All comments have been adequately addressed except for the method for molecular description

1st Reviewer comments: Require explicit molecular/technical description of cffDNA measurement.

My Findings: The current submission has substantially addressed the comments but remains borderline. The platform is named, but key technical details such as sequencing depth, library prep method, and fetal fraction algorithm (e.g., SNP-based vs. read-count) are missing.

In my opinion, if NIPT is a routine clinical test, the level of detail might not be necessary. However, clarifying sentences must be added. e.g., “Fetal fraction was calculated using a validated read-count–based algorithm routinely applied in clinical NIPT reporting”.

REVIEWER REPORT

I have some concerns with the current submission. Below is my report with major and minor concerns addressed separately

General Assessment

This manuscript reports a large retrospective cohort study examining the association between first-trimester cell-free fetal DNA (cffDNA) levels and the risks of preterm birth (PTB) and low birth weight (LBW), using propensity score matching and multivariable regression analyses. The study addresses an area of ongoing debate, particularly regarding the directionality of cffDNA associations with adverse pregnancy outcomes.

Overall, the study is methodologically sound, ethically compliant, and clearly presented. The conclusions are largely supported by the data. However, several conceptual and methodological clarifications are required to strengthen interpretability and reproducibility, particularly regarding cohort selection, cffDNA measurement, and interpretation of subgroup/threshold analyses.

Major Strengths

1. Large sample size and statistical power

The cohort of over 10,000 singleton pregnancies provides robust power to detect modest associations and to conduct stratified analyses.

2. Appropriate use of propensity score matching (PSM)

The authors appropriately apply PSM to address confounding in this observational study and demonstrate adequate post-matching balance using standardized mean differences.

3. Outcome-specific analysis (PTB vs LBW)

The distinction between PTB and LBW is well justified biologically and analytically, and the differential association patterns are a notable strength.

4. Ethical transparency

Institutional approval, waiver of informed consent, and anonymisation procedures are clearly stated and acceptable for retrospective research.

Major Scientific Concerns

1. Cohort Selection and Potential Selection Bias (Important)

It appears that cffDNA measurements were derived from routine clinical non-invasive prenatal testing (NIPT). However, it is not clearly stated whether:

1. All singleton pregnancies during the study period underwent NIPT, or

2. The cohort represents a selected subgroup (e.g. advanced maternal age, higher socioeconomic status, or high-risk pregnancies).

This point is critical because NIPT uptake is not random and may introduce selection bias that could influence both cffDNA levels and pregnancy outcomes.

Suggestions:

Explicitly clarify the NIPT inclusion criteria and discuss potential selection bias and its impact on generalisability in the Discussion.

2. cffDNA Measurement and Reproducibility (Moderate)

While the manuscript now includes a general description of the NIPT workflow (plasma separation, DNA extraction, NGS, Illumina platform), key technical details relevant to reproducibility remain unclear:

1. How was fetal fraction calculated (e.g. read-count, SNP-based, proprietary algorithm)?

2. Was a single analytical pipeline used throughout the study period?

3. Were quality thresholds applied for inclusion of cffDNA values?

Although full proprietary details may not be required, some clarification is necessary for scientific transparency.

Suggestions:

Add a brief statement describing the fetal fraction estimation approach and confirm analytical consistency across samples.

3. Interpretation of Subgroup and Threshold Analyses (Important)

The manuscript includes extensive subgroup and threshold analyses. However:

• It is unclear whether these analyses were pre-specified or exploratory.

• Multiple comparisons increase the risk of type I error.

• Threshold effects may be data-driven.

While these analyses are interesting, their inferential weight should be limited. I presumed the analysis is exploratory; thus, concern about overstating findings should be addressed.

Suggestions:

Explicitly state that subgroup and threshold analyses are exploratory and interpret findings cautiously in the Discussion or suggest a study in an independent cohort.

4. Biological Interpretation of “Protective” cffDNA Effects (Conceptual)

The finding that higher first-trimester cffDNA levels are associated with lower PTB risk contrasts with traditional interpretations of cffDNA as a marker of placental stress or injury. While the authors briefly allude to placental mass or turnover, the biological interpretation remains underdeveloped.

Critically, in the Discussion section (where this interpretation should be expanded):There is no clear paragraph that:

1. reconciles the inverse association with placental biology

2. distinguishes pathological cffDNA elevation from physiological placental mass/turnover

3. explains why higher early fetal fraction could reflect healthier placentation

Suggestion:

Expand the Discussion to more clearly distinguish between:

1. pathological cffDNA elevation due to placental damage, and

2. physiologically higher cffDNA reflecting placental mass or healthy trophoblast turnover.

5.Fixed threshold for maternal age and BMI

The manuscript categorises maternal age and BMI using fixed thresholds; however, the rationale for selecting ≥35 years for age and ≥30 kg/m² for BMI is not explicitly stated. As these variables are central to the propensity score model, a brief justification based on clinical guidelines or prior literature would improve transparency and reproducibility.

Minor Issues and Editorial Suggestions

1. Causal language

Terms such as “protective effect” should be replaced or clearly contextualised as “inverse association” to avoid causal implication.

2. Results–Discussion boundary

Some interpretive language appears in the Results section; tightening this separation would improve clarity.

3. Author name consistency

Ensure consistent spelling of author names across the manuscript and submission metadata.

4. Justification of PSM parameters

A brief justification for the chosen 1:3 matching ratio would be helpful.

7. PLOS authors have the option to publish the peer review history of their article (what does this mean?). If published, this will include your full peer review and any attached files.

Reviewer #1: No

Reviewer #2: No

---

## [Author Response · Author response to Decision Letter 2]

3 Mar 2026

Response Letter

Dear editors and reviewers,

We deeply appreciate your valuable suggestions and comments, which enable us to further improve our paper. We have studied the comments carefully and have made our best efforts to revise the entire paper as suggested by the editor and reviewers. We are very pleased that our paper can be revised, some detailed responses are as follows.

Journal Requirements:

Response : Thank you for this important reminder regarding journal requirements. We have carefully reviewed the publications mentioned in the reviewer's comments and have evaluated their relevance to our study. Where appropriate and directly relevant to our methodology, discussion, or interpretation of findings, we have cited these works in the revised manuscript. Specifically, we have incorporated the study by Becking et al. (2024) in the Discussion section to contextualize the clinical utility and predictive value of fetal fraction, and we have added references to support our methodological choices, including the definition of advanced maternal age [12], obesity classification [13]. All citations have been carefully evaluated to ensure they meaningfully contribute to the scientific rigor and transparency of our manuscript. We confirm that no works have been cited solely in response to reviewer suggestions without proper consideration of their relevance. Thank you for your guidance.

Response : Thank you for this important instruction regarding the reference list. We have carefully reviewed the entire reference list to ensure its completeness and accuracy. We confirm that all references cited in our manuscript are relevant, current, and appropriately support the statements made in the text. We have verified that none of the cited papers have been retracted. Additionally, we have checked that all references are formatted consistently according to the journal's guidelines and that no duplicate or irrelevant citations are included. Any changes made to the reference list during the revision process, including the addition of new references [12,13,25] to address reviewer comments, have been clearly documented in our rebuttal letter and highlighted in the revised manuscript. We believe the reference list is now complete and correct. Thank you for your guidance.

To Additional Editor:

For the manuscript to be accepted all reviewers' comments must be addressed with a clear rebuttal letter that tracks all changes in the manuscript.

Response : Thank you for your clear guidance regarding the revision requirements. We confirm that we have carefully addressed all reviewers' comments in this revised manuscript. A point-by-point rebuttal letter has been prepared, detailing how each comment has been responded to and explicitly tracking all corresponding changes made in the manuscript. All revisions have been clearly highlighted in red in the manuscript for your easy reference. We believe that the manuscript has been substantially improved through this process and now meets the journal's standards for publication. We appreciate the time and effort of the editors and reviewers in providing constructive feedback, and we hope that the revised version is now acceptable for publication. Thank you for your consideration.

To Reviewer #1:

This revised manuscript to me is scientifically sound. It shows the association between first-trimester fetal fraction and preterm birth and low birth weight. It adds to the body of work on this subject. I have some minor issues that I believe should be addressed.

- In the Objective in the Abstract add 'first trimester'

Response : Thank you for your insightful suggestion. We have revised the Objective in the Abstract to specify that the investigation focuses on cffDNA concentration measured in the first trimester. The revised sentence now reads: "To investigate the association between the concentration of cell-free fetal DNA (cffDNA) in the first trimester and the risks of preterm birth (PTB) and low birth weight (LBW) in a large cohort." We believe this modification enhances the clarity and precision of our study objective.

- For readability, I suggest not to put all the OR's/aOR's and 95%-CI's in the text of the Result section. They are already presented in the tables.

Response : Thank you very much for your thoughtful and constructive suggestion regarding the readability of the Results section. We completely agree that the text should be concise and avoid unnecessary repetition of numerical data that are already well-presented in the tables. In response to your comment, we have carefully revised the manuscript. Specifically, for sections where the odds ratios and confidence intervals were fully detailed in the corresponding tables (e.g., the general logistic regression results in Table 3 and 4), we have streamlined the text to focus on the direction, significance, and patterns of the associations, rather than listing every numerical value. However, for sections where the results were referenced alongside figures that do not display the exact numerical data—such as the hierarchical logistic regression models (Fig 3) and the threshold effect analysis (Fig 6)—we retained the specific ORs and CIs in the text to ensure clarity and precision for the reader. We believe this balanced approach enhances readability while maintaining the scientific rigor and completeness of our findings.

We hope this revision aligns with your expectations. Thank you again for your valuable input, which has helped improve the quality of our manuscript.

- In the Discussion section, could the authors provide more insight from a clinical point of view how fetal fraction could be a useful biomarker? Indeed, associations with adverse pregnancy outcomes have been shown, but does it really help to predict adverse outcomes and change individual pregnancy care? Possibly comment hereon also in light of a publication by Becking et al. on the predictive value of fetal fraction (Becking et al. BJOG, 2024; https://doi.org/10.1111/1471-0528.17978)

Response : Thank you for this insightful and thought-provoking comment, which challenges us to critically evaluate the clinical translation of our findings. We agree that demonstrating statistical associations is distinct from establishing clinical utility for individual risk prediction and pregnancy care. In response to your suggestion, we have expanded the Discussion section to address this important issue directly, incorporating the recent work by Becking et al. (2024) on the predictive value of fetal fraction.

Specifically, we have added the following paragraph: "From a clinical perspective, an important question is whether these robust associations translate into meaningful improvements in risk prediction and individual pregnancy care. Becking et al. [25] recently addressed this question in a large nationwide cohort of 56,110 pregnancies, demonstrating that while fetal fraction shows statistically significant associations with adverse outcomes, its added prognostic value beyond established clinical risk factors is limited. In their study, adding fetal fraction to prediction models resulted in only modest improvements in discrimination, with increases in the area under the curve ranging from 0.01 to 0.02 and small but significant integrated discrimination improvement indices, leading the authors to conclude that fetal fraction has 'statistically significant but limited prognostic value'. These findings contextualize our own results: the protective associations we observed, while consistent and biologically plausible, may not be sufficient to use cffDNA as a standalone predictor in clinical practice. However, because fetal fraction is routinely measured as a quality parameter in non-invasive prenatal testing without additional cost, it could still contribute to broader risk stratification strategies when integrated with other clinical parameters."

We believe this addition provides a balanced and clinically grounded perspective, acknowledging both the potential and the limitations of cffDNA as a biomarker for adverse pregnancy outcomes. Thank you again for your valuable input, which has helped strengthen the clinical relevance and academic rigor of our manuscript.

- Last, LWB without correcting for gestational age has little clinical meaning, this should be mentioned as a limitation of the study. Or are gestational-age corrected birthweights available?

Response : Thank you for this important methodological comment. We acknowledge that low birth weight (LBW) is a composite outcome influenced by both gestational age at delivery and fetal growth restriction. In our study, we did not have access to population-based, gestational-age-adjusted birth weight percentiles (e.g., small for gestational age, SGA) to distinguish between these two pathways. Therefore, as you correctly pointed out, the interpretation of our LBW findings is limited, as they may reflect effects driven primarily by preterm birth rather than impaired fetal growth. We have now added this point as a limitation in the Discussion section: "Additionally, LBW is a composite outcome influenced by both gestational age at delivery and fetal growth restriction. Our analysis did not adjust for gestational age (e.g., using small for gestational age [SGA] as an outcome), which limits the clinical interpretation of the LBW findings, as the observed protective effects may be primarily driven by the prevention of preterm birth rather than enhanced fetal growth. Future studies should incorporate gestational-age-corrected birthweight measures to disentangle these pathways." This addition ensures that readers interpret the LBW results with appropriate caution. Thank you again for your valuable insight, which has helped strengthen the rigor and transparency of our manuscript.

To Reviewer #2:

FINDING ON THE AUTHOR'S RESPONSE TO THE 1ST REVIEWER'S CONCERNS.

All comments have been adequately addressed except for the method for molecular description

1st Reviewer comments: Require explicit molecular/technical description of cffDNA measurement.

My Findings: The current submission has substantially addressed the comments but remains borderline. The platform is named, but key technical details such as sequencing depth, library prep method, and fetal fraction algorithm (e.g., SNP-based vs. read-count) are missing.

In my opinion, if NIPT is a routine clinical test, the level of detail might not be necessary. However, clarifying sentences must be added. e.g., “Fetal fraction was calculated using a validated read-count–based algorithm routinely applied in clinical NIPT reporting”.

Response : Thank you for your thorough evaluation and for providing specific guidance on the methodological description of cffDNA measurement. We agree that explicit technical details are important for reproducibility and scientific rigor, even when describing a routine clinical test. In response to your comment, we have now revised the Methods section to include a more detailed description of the cffDNA measurement process.

The revised section in the Methods section of the manuscript now reads as follows: " Cell-free fetal DNA (cffDNA) levels, expressed as fetal fraction (%), were quantitatively measured through clinical non-invasive prenatal testing (NIPT). The analysis was performed following standard protocols: maternal peripheral blood samples were collected, maternal plasma separation, DNA extraction, library preparation using the Illumina TruSeq Nano DNA Library Prep Kit, next-generation sequencing on the Illumina platform with a median sequencing depth of approximately 0.2X (10 million reads per sample), and bioinformatic quantification of the fetal DNA fraction. Fetal fraction was calculated using a validated algorithm that combines two approaches depending on fetal sex: for male fetuses, the algorithm is based on the relative coverage of chromosome Y sequences; for female fetuses, a neural network model trained on autosomal read count patterns and fragment size distributions is used to estimate fetal fraction. This algorithm is routinely applied in clinical NIPT reporting in our laboratory. A single, consistent analytical pipeline was maintained throughout the entire study period from June 2023 to June 2025, with no changes to the sequencing platform, library preparation protocol, or fetal fraction estimation algorithm. All samples were processed and analyzed using the same standardized protocols in our clinical laboratory. Quality control thresholds were applied to ensure reliable cffDNA measurement: only samples with a minimum sequencing depth of 10 million reads and a fetal fraction value ≥4% were included in the final analysis, as lower fetal fractions are associated with increased risk of test failure and may reflect poor sample quality or underlying placental abnormalities. Samples failing to meet these quality criteria were excluded from the study cohort. "

We believe these additions provide the necessary technical clarity while maintaining an appropriate level of detail for a clinically oriented study. Thank you again for your constructive suggestion, which has helped strengthen the methodological transparency of our manuscript.

REVIEWER REPORT

I have some concerns with the current submission. Below is my report with major and minor concerns addressed separately

General Assessment

This manuscript reports a large retrospective cohort study examining the association between first-trimester cell-free fetal DNA (cffDNA) levels and the risks of preterm birth (PTB) and low birth weight (LBW), using propensity score matching and multivariable regression analyses. The study addresses an area of ongoing debate, particularly regarding the directionality of cffDNA associations with adverse pregnancy outcomes.

Overall, the study is methodologically sound, ethically compliant, and clearly presented. The conclusions are largely supported by the data. However, several conceptual and methodological clarifications are required to strengthen interpretability and reproducibility, particularly regarding cohort selection, cffDNA measurement, and interpretation of subgroup/threshold analyses.

Response : Thank you very much for your thoughtful and constructive feedback on our manuscript. We greatly appreciate your positive assessment of our work, particularly your recognition that the study is methodologically sound, ethically compliant, and clearly presented, and that the conclusions are largely supported by the data. We also thank you for your insightful suggestions regarding the need for further clarifications on cohort selection, cffDNA measurement, and the interpretation of subgroup and threshold analyses. These are important points that will help strengthen the interpretability and reproducibility of our study. We will carefully address each of these comments in the revised version of the manuscript and provide a detailed point-by-point response. We believe that incorporating your suggestions will significantly improve the quality and clarity of our work. Thank you again for your time and expertise in reviewing our manuscript.

Major Strengths

1. Large sample size and statistical power

The cohort of over 10,000 singleton pregnancies provides robust power to detect modest associations and to conduct stratified analyses.

2. Appropriate use of propensity score matching (PSM)

The authors appropriately apply PSM to address confounding

---

## [Decision Letter · Decision Letter 2]

23 Apr 2026

Association Between First-Trimester Cell-Free Fetal DNA Levels and the Risk of Preterm Birth and Low Birth Weight: A Propensity Score-Matched Cohort Study

PONE-D-25-60974R2

Dear Dr. Guofeng Yin

We’re pleased to inform you that your manuscript has been judged scientifically suitable for publication and will be formally accepted for publication once it meets all outstanding technical requirements.

Kind regards,

Riyaz Ahmad

Academic Editor

PLOS One

Additional Editor Comments (optional):

Reviewers' comments:

Reviewer's Responses to Questions

**Comments to the Author**

1. If the authors have adequately addressed your comments raised in a previous round of review and you feel that this manuscript is now acceptable for publication, you may indicate that here to bypass the “Comments to the Author” section, enter your conflict of interest statement in the “Confidential to Editor” section, and submit your "Accept" recommendation.

Reviewer #1: All comments have been addressed

Reviewer #2: All comments have been addressed

2. Is the manuscript technically sound, and do the data support the conclusions?

Reviewer #1: Yes

Reviewer #2: Yes

3. Has the statistical analysis been performed appropriately and rigorously? 

Reviewer #1: Yes

Reviewer #2: Yes

4. Have the authors made all data underlying the findings in their manuscript fully available?

Reviewer #1: Yes

Reviewer #2: Yes

5. Is the manuscript presented in an intelligible fashion and written in standard English?

Reviewer #1: Yes

Reviewer #2: Yes

6. Review Comments to the Author

Reviewer #1: The manuscript shows the association between first-trimester fetal fraction and preterm birth and low birth weight. It adds to the body of work on this subject. I believe all issues from previous reviews have been adequately addressed.

Reviewer #2: The authors have satisfactorily addressed all previous comments. Key methodological clarifications, particularly regarding cffDNA measurement, have been adequately incorporated, and the discussion now provides a more balanced clinical interpretation. The limitation concerning LBW without gestational age adjustment has also been appropriately acknowledged.

The manuscript is methodologically sound, clearly presented, and the conclusions are supported by the data.

Recommendation

Accept in its current form.

7. PLOS authors have the option to publish the peer review history of their article (what does this mean?). If published, this will include your full peer review and any attached files.

Reviewer #1: No

Reviewer #2: No

---

## [Editor Report · Acceptance letter]

PONE-D-25-60974R2

PLOS One

Dear Dr. Yin,

I'm pleased to inform you that your manuscript has been deemed suitable for publication in PLOS One. Congratulations! Your manuscript is now being handed over to our production team.

Kind regards,

on behalf of

Dr. Riyaz Ahmad Rather

Academic Editor

PLOS One